# Investigation of Role of Retention Storage in Tanks (Small Water Bodies) on Future Urban Flooding: A Case Study of Chennai City, India

**N. Nithila Devi** [1] **, B. Sridharan** [1] **, V. M. Bindhu** [1] **, B. Narasimhan** [1] **, S. Murty Bhallamudi** [1] **,**
**C. M. Bhatt** [2] **, Tune Usha** [3] **, D. Thirumalai Vasan** [4] **and Soumendra Nath Kuiry** [1,*]

[1]  Environmental and Water Resources Engineering Division, Department of Civil Engineering, IIT Madras, Chennai, Tamil Nadu 600036, India; nithiladevi.n@gmail.com (N.N.D.); krishstee@gmail.com (B.S.); bindhu_v74@yahoo.com (V.M.B.); nbalaji@iitm.ac.in (B.N.); bsm@iitm.ac.in (S.M.B.)

[2]  Disaster Management Studies Department, Indian Institute of Remote Sensing (IIRS), Indian Space Research Organisation (ISRO), Department of Space, Government of India, Dehradun, Uttarakhand 248001, India; cmbhatt@iirs.gov.in

[3]  National Centre for Coastal Research (NCCR), Ministry of Earth Sciences, Government of India, Chennai, Tamil Nadu 600100, India; usha@nccr.gov.in

[4]  Institute of Remote Sensing (IRS), Anna University, Chennai, Tamil Nadu 600025, India; dtvasan@annauniv.edu

*  Correspondence: snkuiry@iitm.ac.in; Tel.: +91-44-2257-4309

**Abstract:** The Adyar River flowing through Chennai Metropolitan Area (CMA) in Southern India functions as a surplus course of upstream water bodies that are locally known as tanks. During northeast monsoons, the river frequently floods the adjoining city areas. In this study, the impact of dredging and disappearance of tanks on flooding in CMA is analyzed under historical, urbanization, and extreme rainfall scenarios utilizing an urbanization-hydrologic-hydraulic modelling framework. The simulated scenarios highlight the importance of the tanks as a flood control measure for CMA. The major conclusions are (a) dredging the tanks uniformly by 2 m can compensate the increase in flooding due to urbanization by 2050 for 1 in 50-year rainfalls and, (b) for disappearance of tanks, 1 in 50-year rainfall can inundate the city akin to 1 in 100-year rainfalls. The study can be useful for making informed decisions on dredging the tanks, land use planning, and flood control measures for the CMA.

**Keywords:** Chennai; tanks; urbanization; dredging; fluvial flooding; flood management

## 1. Introduction

Earth is presently subjected to more frequent natural disasters than ever [1,2]. In the past few decades, a huge fraction of mortalities (nearly 95%) linked with extreme events has been observed in developing nations [1]. Developing nations not only experience higher death rates but also greater economic impacts and are more vulnerable to natural hazards than developed countries [1]. Urban areas in these countries are becoming increasingly vulnerable to frequent and intensive floods [3–5] for a number of reasons: (a) the rise in population due to migration from countryside and other cities [6,7], resulting in extensive unplanned developmental activities and drastic changes in land use; (b) sea-level rise, storm surge, and tropical cyclones under changing climate [5,8–10]; and (c) inefficient flood control measures [11–13]. Heavy rainfall over the Adyar basin in the state of Tamil Nadu, India is a recurring phenomenon during the monsoon seasons due to deep depressions and cyclones originating over the Bay of Bengal. The coastal Chennai city in Adyar basin suffers from frequent inundation during

monsoon seasons due to its low-lying terrain (average elevation is around 6 m). Ill-conceived and poorly maintained storm drains, encroached macro-drainage systems (rivers and canals), and formation of sand bars at the river mouth further escalate this issue. The projected growth of the population of the Chennai Metropolitan Area (CMA) to around 10 million by 2025 [14] is going to further worsen the flooding scenario of the city.

The rapidly growing southern portion of CMA constitutes the lower part of the Adyar basin. The Adyar River flows through CMA for about 12.2 km and falls into the Bay of Bengal. The upper part of the Adyar basin consists of a vast expanse of rural agricultural land and 163 tanks of considerable capacity. The tanks are traditional retention storages that are usually made by damming intermittent streams using crescent-shaped earthen bunds in a cascade down the axes of shallow inland valleys [15–19]. However, the tanks cannot be considered as rainwater harvesting structures that are usually constructed in the residential areas. They exist as a part of the local landscape for several hundred years and are often used for irrigation purposes. The tanks in the catchments vary greatly in size, ranging from relatively small unmanaged water storage systems (e.g., ponds) to large regularly managed water bodies (e.g., lakes and reservoirs) [15–17,19,20]. These tanks play a role like sink-lag-source functions affecting runoff generation [21–26] in the basin. An individual tank can receive water from an upstream catchment and functions as the source by cycling the stored water through evapotranspiration, seepage, over land, and lateral outflow processes. It can also function as a sink by storing the water for longer periods and as a lag by retaining the water temporarily and by releasing it later [27–29]. Therefore, the presence of tanks of considerable storage capacity in the upstream catchments of a city can significantly affect the overall hydrologic response to different drivers such as change in land use and climate. In other words, the tanks in the upper part of a basin function as retention structures maintaining a permanent pool of standing water, with water lost only due to evaporation, seepage, and irrigation releases. The Adyar River flowing through the CMS thus acts as a surplus course for the upstream tanks [30] during heavy rainfall events. As a consequence, runoff from the upstream catchments can collectively impact the peak flood, its timing, and spatial extent in the downstream urban areas [31–34].

A study by [15] reported that around 15% of the storage capacity of the tanks in Adyar basin got lost and that some tanks had lost their storage completely due to heavy siltation. As the land use has drastically changed from agriculture to urban, much of the tanks fell into disuse, was encroached upon, and lost their storage in addition to floodplains of the city's rivers and wetlands [35–37]. Consequently, tanks have either mostly disappeared or cut off from their respective catchments. This, in turn, has impaired the hydraulic functioning of tanks. The poor maintenance and uncontrolled urbanization resulted in significant loss of storage capacities of the tanks, and this has altered the peak flows in the rivers flowing through the city [16,17]. However, conservation of tanks and increasing their capacity through dredging have various direct benefits including improved crop yield, cropping intensity, and groundwater recharge besides flood mitigation [18,38,39]. It was a customary practice in Tamil Nadu, known as "Kudimaramathu" (community maintenance), to desilt local tanks prior to monsoon with a view to get maximum benefit from rainfall [16,17]. In this context, the Tamil Nadu government has recently proposed reviving Kudimaramathu with a budget of US$ 0.14 million [40]. It also consists of removing weeds and deepening of the tanks for proper supply of water for irrigation. The Water Resources Department (WRD) proposed to create additional storage in several tanks by deepening the foreshore area by 1 m to 2 m at an expense of around $16.5 million [41–43]. The decisions were greatly influenced by the disastrous Chennai floods in 2005 and 2015. However, it is important to note here that no prior scientific studies have been carried out to understand the effect of dredging of tanks in the upstream catchments on the flood characteristics of downstream CMA due to its rapid urbanization.

With the tremendous progress in computational resources and high resolution topographical data, the hydrologic-hydraulic modeling framework is being increasingly used for flood prediction in urban catchments [44]. Flooding in Indian cities [45], namely Chennai [30,37,46], Mumbai [47–50], and Pune [51], has been studied extensively. The impact of urbanization on runoff generation was

reported by [30], using historical land use maps of the years 1976 and 2005 in a hydrologic-hydraulic modelling framework for a sub-watershed for Chennai city. It was reported that, for an increase in urban areas by approximately 50%, the flooded areas increased by approximately 15.5%. "What if"-type future flooding scenarios for Chennai city considering urban sprawl, and increased precipitation were reported in [37]. An increase in urban areas by 2.5 times and precipitation by 20% from that of the 2015 flooding is shown to cause an increase in flooded areas by 33%. A flood warning system has been set up [46] for Chennai city very recently by incorporating a regional weather forecasting model, storm surge model, and hydrologic-hydraulic model and recorded flow and stage data. There are several studies that have presented detailed analysis on the utility of small-scale detention tanks that are located within the urban settings in urban flood control and management [52–58]. However, surprisingly, no emphasis has been given on analyzing the influence of tanks alias water bodies that exist as part of the local landscape in the upstream catchments of a city in surface runoff generation and consequently the flooding in the downstream urban areas. This is of greater significance as the tanks in the upper Adyar basin may get encroached or converted to housing or industrial areas due to the rapid expansion of CMA. Encroachment and disappearance of tanks have been cited as major concerns of floods in the CMA by a rapid assessment report [36]. The rainfall pattern over the Adyar basin in the last few decades indicates that rapid urbanization and global warming may be changing the hydro-meteorological condition of the basin [2]. Hence, from the perspective of climate change adaptation and rapid urbanization, it is imperative to understand the role of tanks in flood moderation for various possible historical, urbanization, and climate change scenarios.

It is noteworthy to mention that the sustainability of constructing flood walls, levees, and dredging of the Adyar River is highly arguable due to sewage disposal into the river and the growth of unregulated settlements on its floodplains [13,59,60]. Therefore, as an alternative flood mitigation measure for the basin, this study probes into the need for preserving and increasing the storage capacity of tanks present in rural catchment areas that lie upstream of the CMA. This is a source-control flood mitigation measure that can reduce the runoff rate and severity of a flood. However, periodic dredging of tanks, especially after the occurrence of extreme floods, is crucial. The objective of the present study is thus set to analyze the role of tanks and the effectiveness of dredging them in altering flooding characteristics of CMA for three different scenarios: (i) historical rainfall and present urban condition, (ii) historical rainfall and future urbanization, and (iii) combined future extreme rainfall and future urbanization. Results from this study will help in gain a scientific understanding of the measures that may be planned by the government for an effective flood control and climate change adaptation strategies for crucial water resources.

## 2. Methodology

Two historical flood events are considered for the analysis herein, and more details on the flood events can be found in [42,61–64]. It was found from the frequency analysis of the Meenambakkam rain gauge within the study area that the 2005 and 2015 floods roughly correspond to 50-year and 100-year return period rainfall events, respectively. Therefore, the impact of dredging of tanks is analyzed by simulating the moderate 2005 and extreme 2015 flood events under historical, future urban sprawl, and climate change scenarios. The framework (Figure 1) consists of the following components: (i) scenarios definition, (ii) frequency analysis of NASA Earth Exchange Global Daily Downscaled Projections (NEX-GDDP) data, (iii) prediction of urbanization, (iv) hydrologic modelling, (v) hydraulic modelling, and (vi) impact analysis of the simulated flood scenarios.

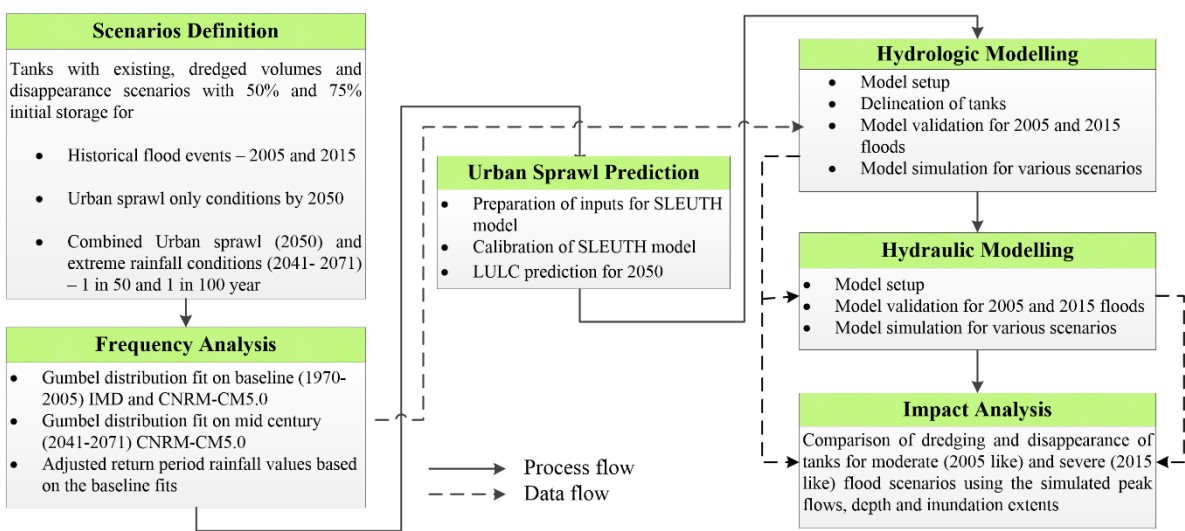

**Figure 1.** Flow chart explaining the components involved in the study.

## 2.1. Study Area

The Adyar basin, chosen as the study site, occupies an area of about 724 km² (Figure 2). As mentioned earlier, the Adyar River originates as a surplus course from water bodies in the upper part of the basin and flows through the city before falling into Bay of Bengal. The total length of the river is about 43 km. The river remains dry for most of the year with flows largely occurring during the North-East (NE) monsoon (October to December). Chennai International Airport is located by the Adyar River, and its secondary runway is across the river. Many industries, business centers, hospitals, schools, and residential colonies are located on either side of the river. The CMA being a business hub has witnessed steep growth in the manufacturing, health care, retail, and IT sectors in the past few decades [65]. The city is the fourth largest metro in India, and it has become a center for tremendous economic activities. On the downside, the city is experiencing rapid and unplanned expansion due to migration of population from various parts of the country. This has resulted in significant increase in impervious areas and encroachment of water bodies, riverbanks, drains, and marsh lands. Chennai being a coastal city, lying on the thermal equator, has a tropical hot and humid climate. The average annual rainfall of CMA is around 1400 mm. During the NE monsoon, the city receives most of its annual rainfall (approximately 800 mm). Therefore, the city is in a critical position to harvest this seasonal rainfall using its water bodies and reservoirs to meet the increasing water demand. As mentioned earlier, the deep depressions and cyclones that develop over the Bay of Bengal during the NE monsoon period result in frequent flooding of the city. A 15–60 m wide Buckingham canal and some major and minor drains aid in draining the flood waters in the city into the sea. However, the sewage outfalls, bridges, and metro stations disrupt the flow in Buckingham canal and its width is restricted at many places to 10 m [13]. The Mambalam drain that lies in the center of the city draining storm water into the river is also restricted at many places due to sewage outfalls and urbanization. The upper portion of the Mambalam drain is largely the remnants of a disappeared water body. According to [13], the places in and around the Mambalam drain, namely T. Nagar, Nandhanam, etc., were the 2005 flood hotspots. Additionally, the urbanized R.A. Puram area located near the Adyar creek is susceptible to flooding due to the low-lying terrain.

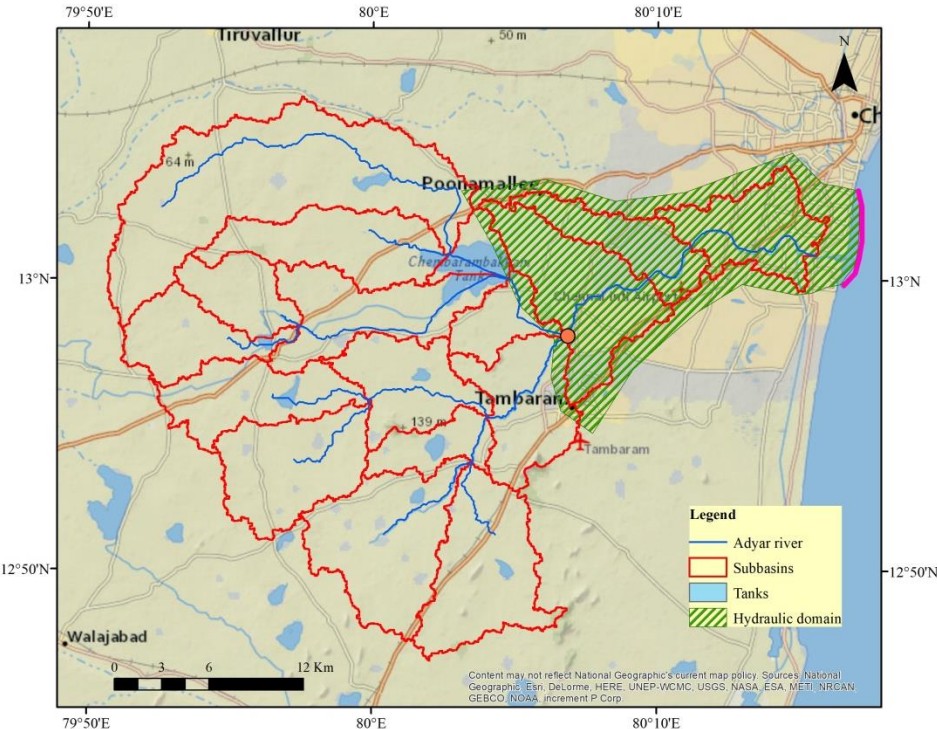

**Figure 2.** Study area map: the orange dot in the figure indicates the location where upstream boundary condition is specified, and the pink line indicates the location of downstream boundary condition.

*2.2. Scenarios Definition*

The initial storage in the tanks at the onset of the extreme rainfall event dictates the role of tanks in flood moderation. Based on volume estimates of the tanks and simulations of hydrological model (refer to Section 2.5 for details on the model set up), it is found that a rainfall between 75 mm and 100 mm is good enough to fill the tanks at the most by 50% and 75% of its capacity, respectively, in the rural portion of the watershed. The NE monsoon rainfall (NEMR) exhibits both spatial and temporal vagaries in the basin. A study by [66] presented preliminary results that suggest considerable intra-seasonal variability of NEMR. It has to be noted that the occurrence of extreme rainfall events is limited. This in turn makes it difficult to analyze the wet spells preceding the extreme events. Consequently, with a view to address the uncertainty in assuming the initial storage of the tanks, 50% and 75% initial storages in the tanks were considered for various historical, urban sprawl, and climate change scenarios. The study considers not only the existing volume of tanks but also the increased volumes of dredged tanks for various scenarios. As mentioned earlier, based on the proposal of the WRD of Tamil Nadu [41–43], the dredging of the tanks by 1-m and 2-m scenarios are considered in the study. The scenarios for disappearance of tanks are included only in future simulations as the tanks might lose the storage completely with time due to poor maintenance and management.

Tables 1 and 2 give a detailed description of the scenarios that are used in analyzing the effectiveness of the proposed flood management strategies. The notations used for depicting the scenarios comprise two parts: one to denote the Rainfall/LULC and the other to indicate tank-related conditions. The two parts listed in Table 1 in the last two columns can be combined to give various scenarios that are used in this study. For example, US100-RCP4.5-TP2m-F50 represents the scenario with urban sprawl corresponding to the year 2050 and 100-year 24-h duration return-period rainfall derived from CNRM-RCP 4.5 for the period (2041–2071). All 163 tanks presently available are dredged by 2 m uniformly throughout the water spread area, the dredged tanks being filled up by 50% at the beginning of the simulation.

**Table 1.** Definition of scenarios.

| Rainfall | Category of the Scenarios | Notation | |
|---|---|---|---|
| | | **Rainfall/LULC** | **Tank Conditions** |
| 2005 and 2015 floods | Historical—the land use land cover (LULC) and the rainfall pertaining to 2005 and 2015 floods are considered. | H05/H15 | TP-F50 TP-F75 |
| | Future urban sprawl—the LULC corresponding to 2050 and the rainfall pertaining to 2005 and 2015 floods are considered. | US05/US15 | TD TP-F50 TP-F75 TP1m-F50 TP1m-F75 TP2m-F50 TP2m-F75 |
| 24 h–50- and 100-year return period floods | Baseline—IMD rainfall in the period 1970–2005 corresponding to the same frequency of occurrence as that of the heaviest one-day rainfall of the 2005 and 2015 events are considered. | BL50/ BL100 | TP-F50 TP-F75 |
| | Midcentury projections—urban sprawl corresponding to 2050 and NEX-GDDP rainfall in the period 2041–2071 corresponding to the same frequency of occurrence as that of heaviest one day rainfall of the 2005 and 2015 events are considered. | US50-RCP4.5/ US100-RCP4.5/ US50-RCP8.5/ US100-RCP8.5 | TP-F50 TP-F75 TP1m-F50 TP1m-F75 TP2m-F50 TP2m-F75 TD |

**Table 2.** Description of scenarios.

| Notation | Description |
|---|---|
| H05 | Represents historical land use and heavy rainfall event of the year 2005. |
| H15 | Indicates historical land use and heavy rainfall event of the year 2015. |
| US05 | Represents urban sprawl scenario for the year 2050 and heavy rainfall event of the year 2005. |
| US15 | Denotes urban sprawl scenario for the year 2050 and heavy rainfall event of the year 2015. |
| BL50 | Used to represent base line one-day IMD rainfall of a 50-year recurrence interval that is derived for the period 1970–2005. LULC of the year 2015 is used. |
| BL100 | Used to represent base line one-day IMD rainfall of a 100-year recurrence interval that is derived for the period 1970–2005. LULC of the year 2015 is used. |
| US50-RCP4.5 | Points to the scenario that uses the urban sprawl scenario for the year 2050 and CNRM-RCP 4.5 rainfall of the 50-year recurrence interval that is derived for the period 2041–2071. |
| US100-RCP4.5 | Denotes the scenario that uses urban sprawl for the year 2050 and CNRM-RCP 4.5 rainfall of 100-year recurrence interval that is derived for the period 2041–2071. |
| US50-RCP8.5 | Represents the scenario that uses urban sprawl for the year 2050 and CNRM-RCP 8.5 rainfall of the 50-year recurrence interval that is derived for the period 2041–2071. |
| US100-RCP8.5 | Points to the scenario that uses urban sprawl for the year 2050 and CNRM-RCP 8.5 rainfall of the 100-year recurrence interval that is derived for the period 2041–2071. |
| TP | Existing volumes of the 163 tanks that are considered for the study are included in the hydrologic model. These scenarios represent the baseline conditions or the tanks' present conditions. |
| TP1m | The 163 tanks are deepened by 1 m uniformly throughout the water spread area, and the increased volumes are included in the hydrologic model. |
| TP2m | The 163 tanks are deepened by 2 m uniformly throughout the water spread area, and the increased volumes are included in the hydrologic model. |
| F50 | Prestorage of 50% of the total storage capacity of the tanks. |
| F75 | Prestorage of 75% of the total storage capacity of the tanks. |
| TD | This scenario considers the absence of tanks or tank disappearance in the study area. In other words, the depression storage is considered 0 in the study area. |

### 2.3. Frequency Analysis of NEX-GDDP Data

The rainfall for the midcentury (2041–2071) period in this study uses daily time series of rainfall obtained from NASA Earth Exchange Global Daily Downscaled Projections (NEX-GDDP) datasets. The NEX-GDDP dataset includes downscaled climate scenarios obtained from General Circulation Model (GCM) runs that were carried out under the Coupled Model Intercomparison Project Phase 5 (CMIP5) for two greenhouse gas emission scenarios called Representative Concentration Pathways (RCPs) 4.5 and 8.5 at a spatial resolution of 0.25° (approximately 25 km × 25 km). Studies by [67,68] pointed out that CNRM-CM5.0 (Centre National de Recherches Meteorologiques Coupled Global Climate Model, Version 5) and GFDL-CM3.0 (Geophysical Fluid Dynamics Laboratory Climate Model, Version 5) CMIP 5 models can realistically simulate the June–September summer monsoon rainfall over India. However, to limit the length of the paper, midcentury (2041–2071) rainfall data corresponding only to the CNRM-CM5.0 model is used. The study area extends across four CNRM-CM5.0 grids that are represented in this paper as grids 1, 2, 3, and 4 for convenience. Grids 1, 2, 3, and 4 are centered at the coordinates (79.875° E, 12.875° N), (80.125° E, 12.875° N), (79.875° E, 13.125° N), and (80.125° E, 13.125° N), respectively. The annual maximum daily rainfall series is obtained for the period 1970–2071 for all CNRM-CM5.0 model grids. Gumbel (extreme value type I) distribution is then used to estimate the rainfall depths corresponding to various recurrence intervals/return periods. In an attempt to correct the bias in CNRM-CM5.0 data, the frequency analysis is repeated for the Indian Meteorological Department (IMD) grid data available at the resolution 0.25° × 0.25°. IMD data has a similar resolution to that of the CNRM grids for the baseline period (1970–2005). For each return period and grid, the ratio of the corresponding IMD to CNRM-CM5.0 rainfall depths is obtained for the baseline period. The rainfall depth ratios for various return periods are then multiplied with the corresponding CNRM-CM5.0 midcentury rainfall depths to get the bias-adjusted values. Figure 3 shows the relationship between bias-adjusted rainfall depths for different return periods corresponding to CNRM-CM5.0 datasets for the RCP 4.5 and 8.5 scenarios.

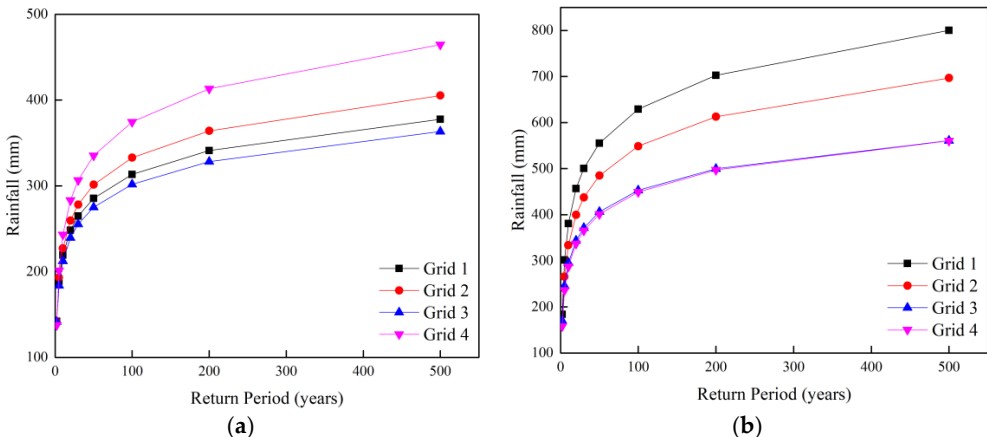

**Figure 3.** Bias-adjusted 24-h rainfall (mm) at various recurrence intervals for the CNRM-CM5.0 model: (**a**) RCP 4.5 and (**b**) RCP 8.5 models.

### 2.4. Urban Sprawl Prediction

In this study, the SLEUTH model [69–71] is used for urban sprawl prediction of Chennai city by 2050. "SLEUTH" is an acronym based on the inputs used in the model: slope, land use, exclusion, urban, transport, and hill shade. It is a probabilistic cellular automata model, which is widely used for urban growth prediction. The model requires urban extents corresponding to four years, while two land use maps are required: one at the beginning and the other at the end of the calibration period. In this study, these maps are prepared by performing unsupervised classification on Landsat imageries belonging to 28 October 2000, 28 June 2002, 9 March 2011, and 25 March 2017. The images are considered from the year 2000, when Chennai city started growing rapidly. The intervals between the images

are well represented to capture the dynamics of urban sprawl over two decades. The slope and hill shade maps are obtained from the Shuttle Radar Topography Mission (SRTM) DEM of 30 m × 30 m resolution. The transport network data is obtained from Open Street Map (OSM) repository.

The basic processing units of the model are termed cells. The cells get new states for each time step based on the transition rules applied to the previous states of the cell and its neighbors. The transition rules that influence the probability of urbanization are determined based on growth parameters that are to be calibrated [72,73]. SLEUTH uses brute force calibration, and in order to mimic the random processes associated with urban growth, Monte-Carlo (MC) simulations are used [74]. The results of the calibration run can be interpreted with the help of fit statistics, such as compare metric and the Lee–Sallee index. The compare statistics is the ratio of the number of modelled populations of urban pixels to the observed populations for the final control year [75]. The Lee–Sallee shape index is the ratio of intersection and union of the simulated and observed urban areas, averaged over all control years [72,76]. Many studies have used Lee–Salle metric to narrow down the coefficient space [73,77–81]. Apart from the aforementioned spatial metrics, the SLEUTH model also estimates non-spatial pattern metrics, namely clusters and edges metrics. Clusters and edges metrics are ordinary least-squares regression scores for the modelled and actual urban clusters and edge pixels, respectively [75]. For the Adyar basin, the final calibration phase resulted in values of 0.6, 0.61, 0.88, and 0.7 for the compare, Lee-Sallee, clusters, and edges metrics, respectively. It can be found from literature that the obtained values of the metrics are within the ranges as reported in similar studies [77–79,81]. The calibrated values of the diffusion, breed, spread, slope, and road coefficients are 68, 67, 25, 25, and 44, respectively. In the prediction mode, the urban extent is simulated for every year in the period 2000–2050. The predicted LULC maps for the years 2005, 2015, and 2050 are shown in Figure 4. Table 3 presents the distribution of predicted LULC classes for the years 2005, 2015, and 2050. It can be seen from Table 3 that % of urban pixels increased from 9.6% to 45.9% in the period 2005–2050.

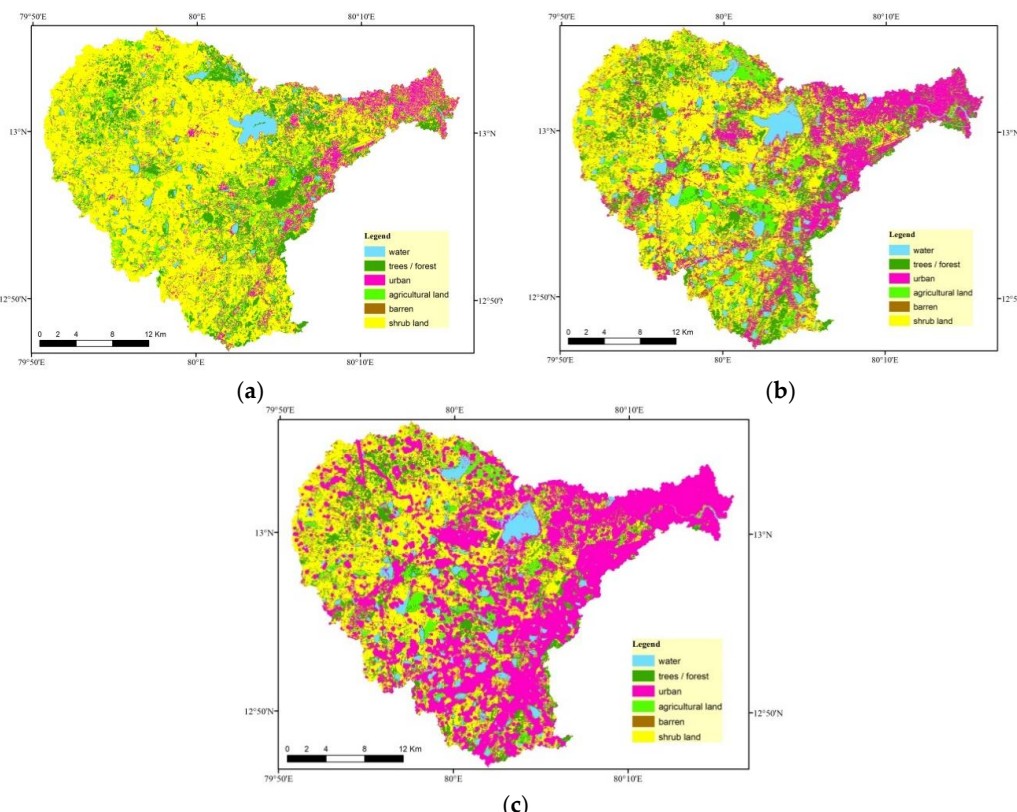

**Figure 4.** Land Use Land Cover (LULC) maps for (**a**) 2005, (**b**) 2015, and (**c**) 2050.

**Table 3.** Percentage distribution of different LULC classes in the Adyar basin.

| LULC | 2005 | 2015 | 2050 |
|---|---|---|---|
| Water | 3.2 | 6.9 | 5.3 |
| Trees/Forest | 17.2 | 19.5 | 12.0 |
| Urban | 9.6 | 20.3 | 45.9 |
| Agriculture | 8.8 | 6.6 | 4.1 |
| Barren | 0.1 | 2.6 | 1.6 |
| Shrub | 61.0 | 44.2 | 31.1 |

*2.5. Hydrological Modelling*

The HEC-HMS (Hydrologic Engineering Center-Hydrologic Modeling System) model [82] developed by U.S. Army Corps of Engineers is used in this study to generate flood hydrographs at the inlet of the hydraulic domain for the scenarios in Table 1. HEC-HMS requires DEM, soil type, LULC, and rainfall data as inputs. An SRTM DEM of 30 m × 30 m resolution is used for delineating the sub-basins of the study area (Figure 2). The Curve Number (CN) method is used to estimate the effective rainfall [83]. The CN method requires soil and LULC information as inputs. The European Soil Data Centre (ESDAC) [84] soil repository and LULC maps (refer to Section 2.4) for 2005, 2015, and 2050 are used for this purpose. The meteorological model includes four rain gauges for the 2015 flood and five rain gauges for the 2005 flood that are spread across the basin. Inverse distance method is used to interpolate the precipitation data throughout the basin. For the 2015 flood event, the hourly rainfall data is available from the rain gauges [36,37], while for the 2005 flood event, the daily accumulated rainfall that was reported in [30] is distributed into 3-hourly data using the TRMM (Tropical Rainfall Measuring Mission) data. For the future climate change scenario, the adjusted extreme rainfall data for the period 2041–2071 is obtained from CNRM-CM5.0 model (refer to Section 2.3). Similarly, the adjusted mid-century 24-h rainfall values corresponding to 50- and 100-year return periods are distributed into 3-hourly data using the TRMM data of the 2005 and 2015 flood events, respectively. Kinematic wave routing is chosen to route the channel flow through the basin.

Around 163 tanks are identified and delineated in the upper part of the Adyar basin with the aid of Google Earth image repository. The LIDAR DEM of 2 m × 2 m spatial resolution is used to calculate water spread area and volume of the tanks. The minimum, mean, and maximum water spread areas of the tanks are estimated to be 0.003, 0.41, and 5.25 km$^2$, respectively. The sum of the volume of tanks falling under each sub-basin is entered as depression storage in the hydrologic model. For the "TP1m" and "TP2m" dredging scenarios, the DEM values within the water spread areas of the tanks are reduced by 1 m and 2 m, respectively; the corresponding increased volumes are summed up sub-basin-wise; and depression storages are estimated accordingly. Therefore, different dredging scenarios are defined by varying the depression storage for each sub-basin. As initial water levels in the 163 tanks are not known for the 2005 and 2015 rainfall events, the percentage of initial storage in the tanks are varied as 50% and 75% of the tanks' original volumes in the HEC-HMS setup, which are represented as "F50" and "F75" scenarios, respectively, in Table 1. Eventually, the runoff hydrographs that are upstream boundary conditions for the hydraulic model are generated for all the scenarios. While for the "US" scenarios, the CN-related parameters are updated by using the simulated urban sprawl by 2050.

Validation of Hydrological Model

The hydrological model that is set up for the historical 2005 and 2015 flood events are validated separately. As the Adyar basin is not well instrumented, the observed data is extremely limited. Hence, the observed flow data for calibration is limited to the reservoir inflow data estimated from the measured release over the spillways and change in observed storage. Moreover, there was a significant downpour before the isolated peaks in 2005 and 2015 rainfall events. Hence, to be on the conservative side and to account for the water released from the tanks, 75% initial storage in the tanks for historical

2005 and 2015 flood scenarios are used as antecedent conditions for validation against the recorded inflows into the Chembarambakkam reservoir (Figure 5) located in the upper part of the Adyar basin. Upon comparison with the observed inflow values, the coefficient of regression ($R^2$) values of 0.97 and 0.92 are obtained for the 2005 and 2015 simulations, respectively. The simulated peak flow for the historical floods in 2005 and 2015 defined as the existing scenarios with 75% initial storage in the tanks are found to be 2785 $m^3/s$ and 4230 $m^3/s$, respectively, at the outlet of the basin. The peak discharge value during the 2015 flood event is found to be significantly higher than the flow carrying capacity (2038 $m^3/s$) [64] of the Adyar River.

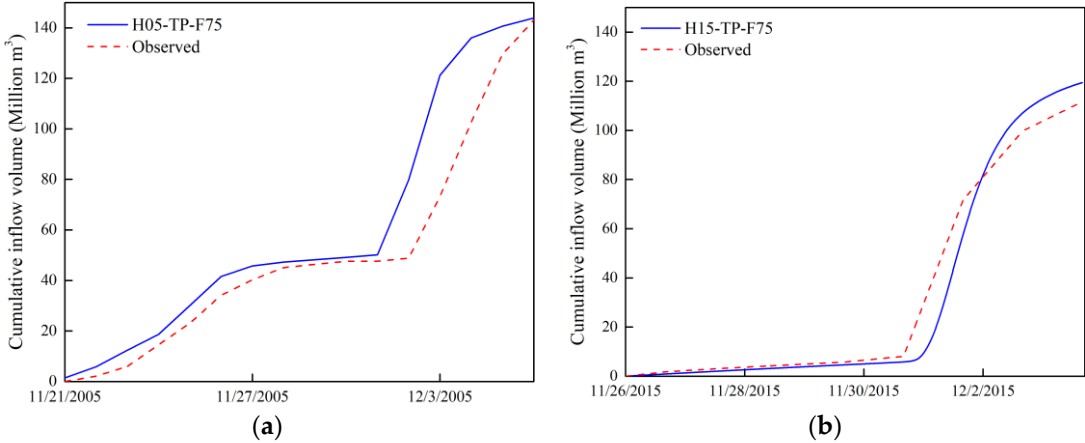

**Figure 5.** Observed and simulated cumulative inflow volume into the Chembarambakkam reservoir for the (**a**) 2005 (coefficient of determination, $R^2$ = 0.92) and (**b**) 2015 flood events ($R^2$ = 0.97).

### 2.6. Hydraulic Modelling

The study [30] simulated different return period floods with 1976 and 2005 land use patterns using the HEC-RAS (Hydrologic Engineering Center—River Analysis System) 1D model developed by U. S. Army Corps of Engineers to highlight the negative impact of urbanization on flooding in CMA. However, it has to be noted that a 1D model cannot simulate the 2015 flood event with high accuracy due to its widespread inundation. The study [37] simulated the 2015 Chennai flood event using HEC-RAS 2D version 5.0 [85] and SRTM DEM of the resolution 30 m. Such a coarse resolution DEM cannot accurately represent the bathymetry of important topographical features such as the Adyar River in the CMA (approximately 80–200 m wide), the Buckingham canal (approximately 15–60 m wide), and the Mambalam drain (approximately 15 m wide). Therefore, a LIDAR DEM of a high spatial resolution of 2 m × 2 m that better represents the smaller topographical features of CMA is used to set up the hydraulic domain in this study. For the ocean side, the publicly available General Bathymetric Chart of the Oceans (GEBCO) gridded data is used. As the maximum number of grids that are allowed in the HEC-RAS 2D model is approximately 350,000, computation is not feasible on such a high-resolution grid. Therefore, the HEC-RAS sub-grid modelling is utilized herein with a computational grid of 30 m resolution together with 2 m resolution LIDAR DEM for topography to capture the detailed hydrodynamics. The sub-grid concept helps in achieving high accuracy by considering minute topographical features in lesser simulation time [86]. The flood hydrographs obtained at the river inlet from the hydrologic model are prescribed as inflow boundary conditions to the hydraulic model. In the absence of observed tidal gauge data at the mouth of the Adyar River during the flood events, predicted tide data from WXTide software (available at http://www.wxtide32.com/index.html) is supplied as the downstream boundary condition. The WXTide software was used by [87] to assess the vulnerability of the Chennai coast. According to [88], high sea levels are observed in the month of November, during which the city usually receives the maximum amount of rainfall. The predicted tidal data exhibited a similar trend and is in good agreement with the observed data as reported by [89]. For simplicity,

a single value of the Manning's coefficient of 0.035 m$^{-1/3}$s is used for bed roughness as suggested in [37] for the same hydraulic domain.

Validation of Hydraulic Model

The simulated maximum inundation extent for 2015 flood is shown in Figure 6. There is no observed inundation extent available for fluvial flooding separately as the CMA also suffered from pluvial flooding due to its low-lying terrain, urbanization, encroachment of water bodies, and poor storm water management practices. Hence, the simulated inundation extents are qualitatively compared with those reported in [37,46]. As mentioned earlier, the simulated extent in [37] is obtained using SRTM 30 m DEM, whereas the inundation extent in [46] is obtained using the same 2 m high-resolution LIDAR DEM used in this study. Therefore, inundation along the smaller topographical features like Mambalam drain, Buckingham canal, and R.A Puram area closer to the Adyar creek is better captured in this study (Figure 6), similar to that reported in [46]. It needs to be emphasized that the use of high-resolution DEM and the concept of sub-grid helps to capture the flood dynamics more realistically. However, the authors of [37] could not capture such flow dynamics due to coarse resolution DEM. Overall, it is found that the simulated inundation extent for the 2015 flood event is as accurate as [46] and better than [37]. There is no inundation map available for the 2005 flood event. However, it can be considered that the simulated inundation extent for the 2005 flood event is reasonably accurate as the same hydraulic model setup is used with the corresponding boundary conditions.

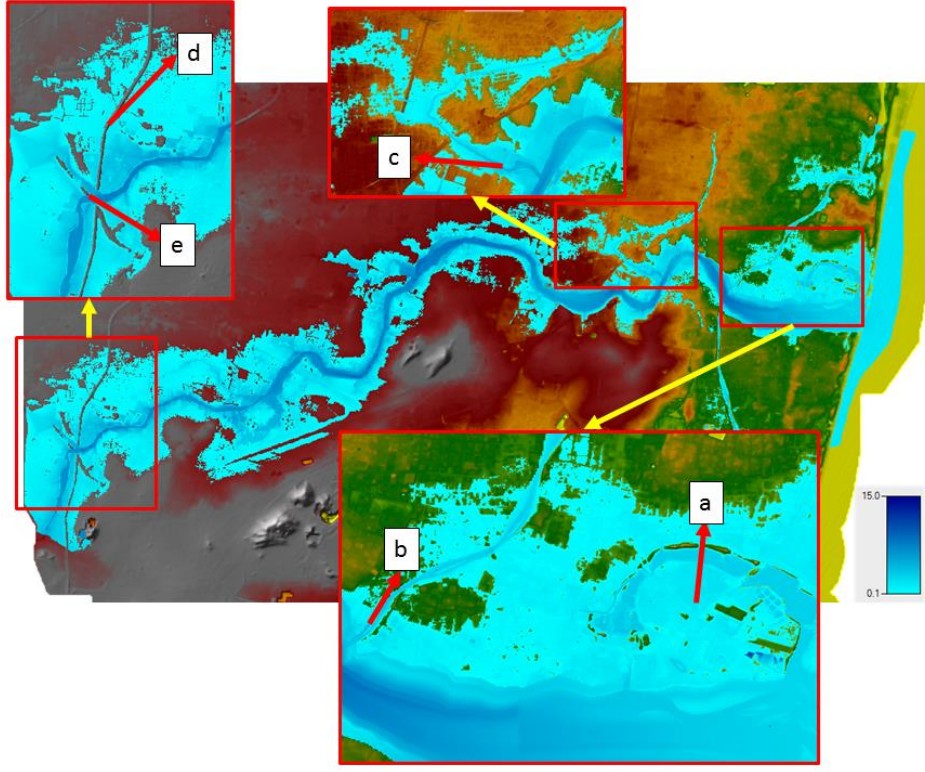

**Figure 6.** Simulated maximum inundation extent for the H15-TP-F75 scenario as viewed in RAS-Mapper: the zoomed in portions are (**a**) R.A. Puram, (**b**) Buckingham canal, (**c**) Mambalam drain, (**d**) Chennai Bypass road, and (**e**) Pallavaram–Kundrathur road.

In addition to the inundation area, the hydraulic model is also validated with observed flood depths. Surveyed watermarks that were collected immediately after the 2015 flood event are available at various locations within the study area through field campaigns by different agencies such as the National Remote Sensing Centre, Institute for Remote Sensing—Anna University, National Centre for Coastal Research, and Indian Institute of Technology Madras [37]. As the study focuses on

only fluvial flooding within the hydraulic domain, validation points that are closer to the river, its macro-drains, and canals are considered (Figure 7a). Upon comparison of simulated flood depths with the observed depths, $R^2 = 0.81$ is obtained for the 75% initial tank storage scenario (Figure 7b). For the 2005 flood event, the maximum flood depths are available only at Maraimalai Adigal and Thiru vi ka bridges, as reported in [13]. The simulated depths (5.6 and 9.4 m) are similar to the observed depths (5.4 and 9 m).

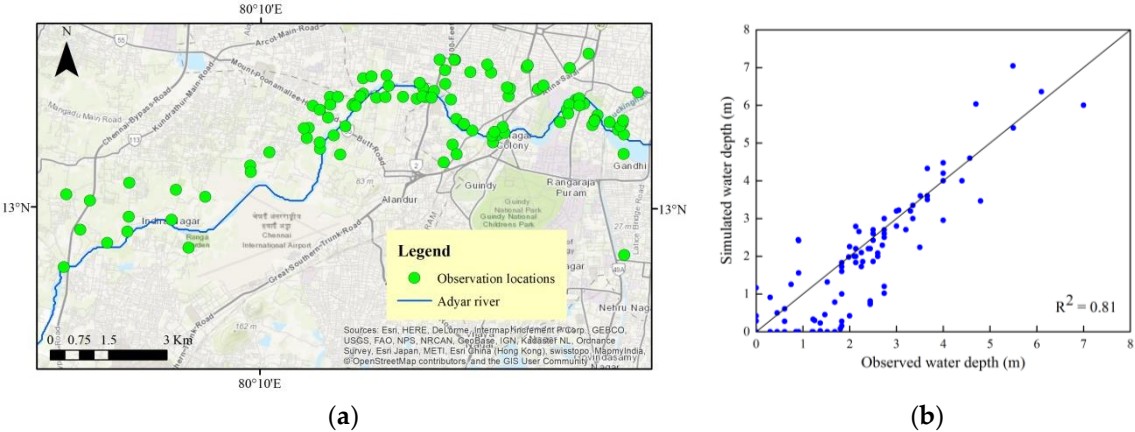

(**a**)                                                                          (**b**)

**Figure 7.** Validation of the 2015 flood event: (**a**) locations of the surveyed maximum watermarks and (**b**) scatter plot of simulated vs. observed inundation depths for 75% initial storage in all the tanks.

## 3. Results and Discussion

### 3.1. Hydrological Results

Various aspects of flood management regarding tanks like initial storage, dredging, and disappearance are discussed in detail.

#### 3.1.1. Impact of Initial Storage in Tanks

The upstream flood hydrographs for various scenarios are compared in this section in Figures 8–14. It can be seen from Figures 8–14 that the decrease in peak flood discharge in urban sprawl-tank dredging scenarios is more pronounced only when the initial storage in the tank is less than 50% prior to the occurrence of flood. For the US05-TP2m dredging case, the peak discharge in a 2005-like flood will be closer to that of the historical flood itself if 50% of storage is available in all the tanks prior to occurrence of the event. However, such a significant reduction is not observed when 75% of the storage is already filled prior to occurrence of the event. Table 4 shows relative change in peak flows with respect to the corresponding US or US-RCP cases for different conditions of dredging and filling of tanks prior to occurrence of events. It can be seen from Table 4 that, as far as adaptation measures to counter impacts of future urbanization and climate change are concerned, the effect of initial storage on the reduction of flood peak is more pronounced when tanks are dredged by 2 m as compared to dredging them by 1 m. For example, in the US50-RCP4.5–50-year return period rainfall events, for 50% initial storage in the tanks, the percentage decrease of flood peaks are 3.7 and 4.4 times greater than that for the 75% initial storage in the tanks in the cases of the 1-m and 2-m dredging scenarios, respectively. Hence, it can be said that the degree of flood moderation will vary depending on the initial storage in tanks. In addition, the effectiveness of dredging of tanks in flood control is also dependent on the initial storage or the volume available in the tanks for harvesting the rainfall.

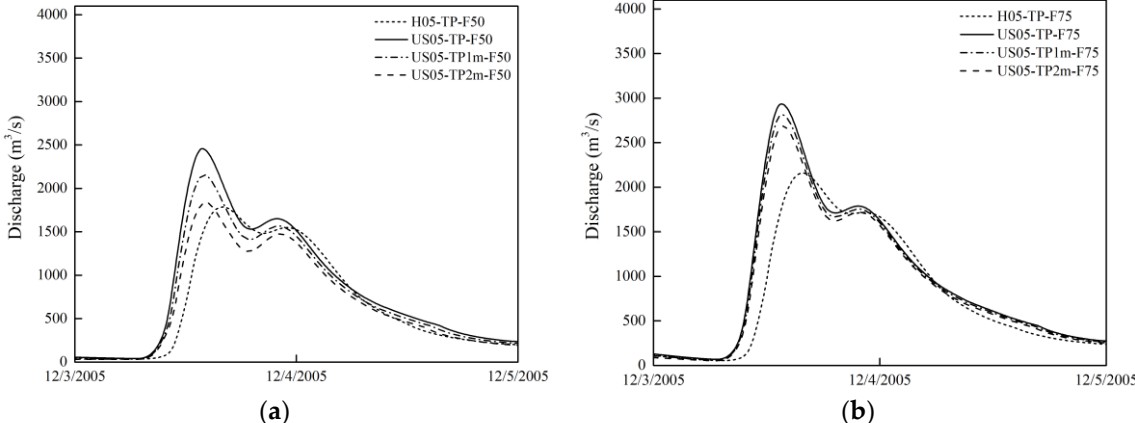

**Figure 8.** Upstream hydrographs for 2005 flood scenarios corresponding to (**a**) 50% and (**b**) 75% initial storage in the tanks.

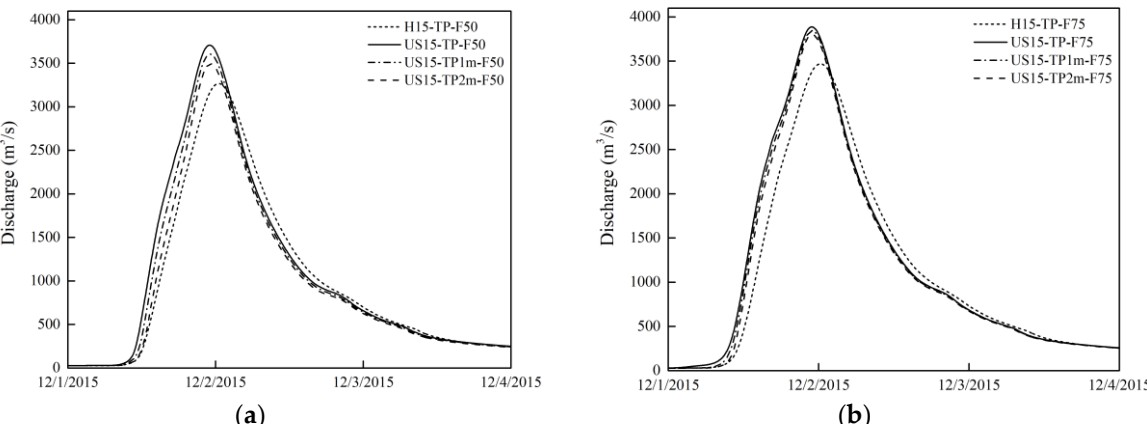

**Figure 9.** Upstream hydrographs for 2015 flood scenarios corresponding to (**a**) 50% and (**b**) 75% initial storage in the tanks.

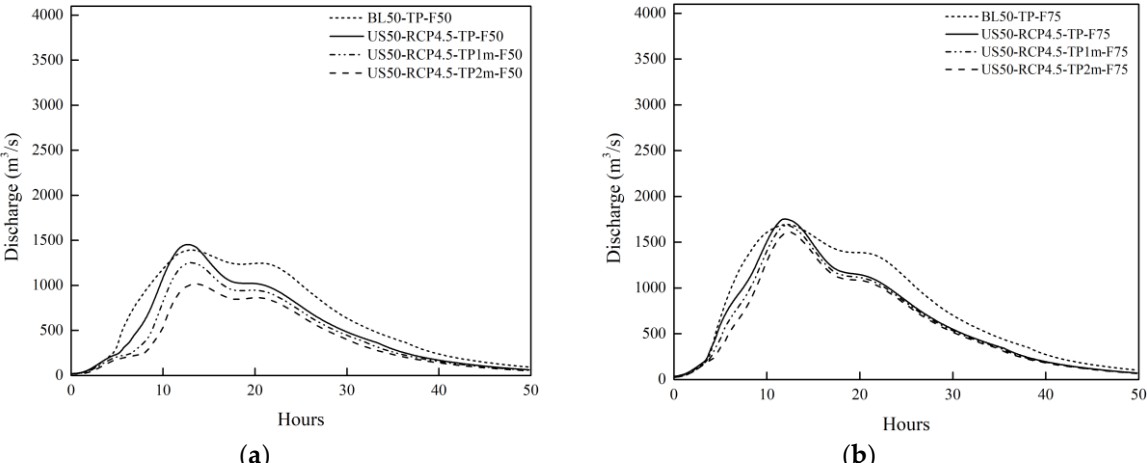

**Figure 10.** Upstream hydrographs for the 50-year return period CNRM 4.5 rainfall flood scenarios corresponding to (**a**) 50% and (**b**) 75% initial storage in the tanks.

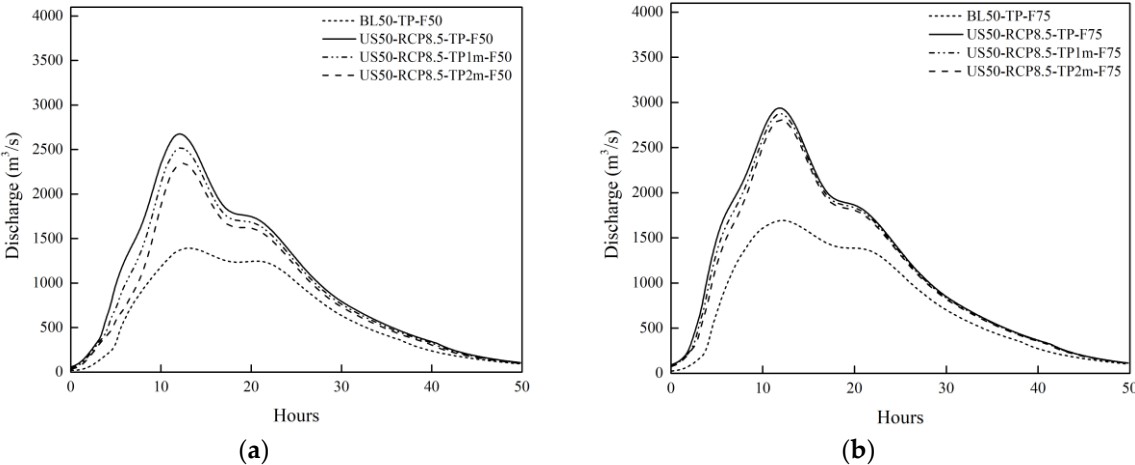

**Figure 11.** Upstream hydrographs for the 50-year return period CNRM 8.5 rainfall flood scenarios corresponding to (**a**) 50% and (**b**) 75% initial storage in the tanks.

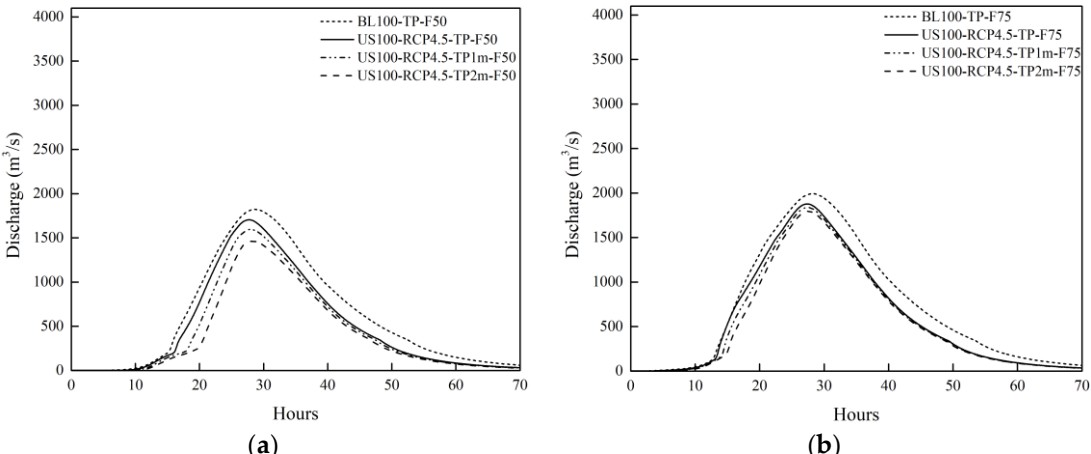

**Figure 12.** Upstream hydrographs for the 100-year return period CNRM 4.5 rainfall flood scenarios corresponding to (**a**) 50% and (**b**) 75% initial storage in the tanks.

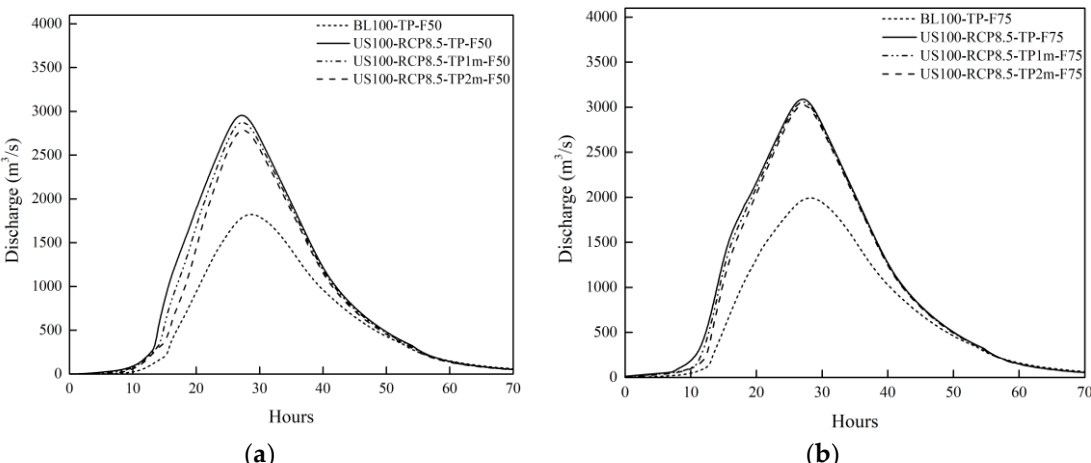

**Figure 13.** Upstream hydrographs for the 100-year return period CNRM 8.5 rainfall flood scenarios corresponding to (**a**) 50% and (**b**) 75% initial storage in the tanks.

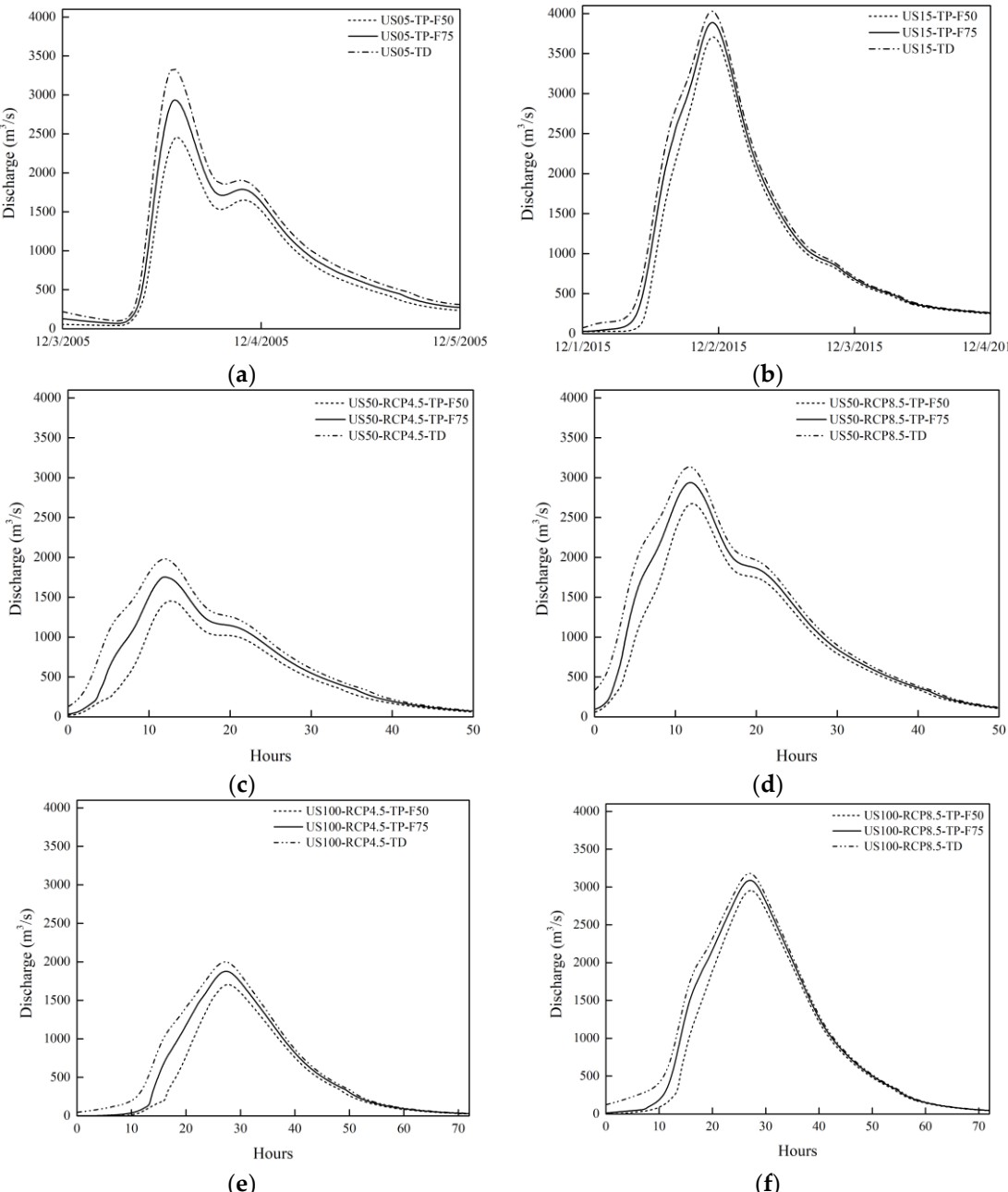

**Figure 14.** Upstream hydrographs including tanks disappearing scenarios for urban sprawl: (**a**) 2005 and (**b**) 2015 50-year return period—(**c**) CNRM 4.5 and (**d**) CNRM 8.5—and 100-year return period—(**e**) CNRM 4.5 and (**f**) CNRM 8.5—flood events.

**Table 4.** Relative percentage change in peak flows for various scenarios in comparison to the corresponding existing scenarios of urban sprawl and climate change.

| Rainfall | Scenarios | TP1m | | TP2m | | TD | |
|---|---|---|---|---|---|---|---|
| | | **F50** | **F75** | **F50** | **F75** | **F50** | **F75** |
| 2005 | US05 | −12.1 | −4.1 | −28.6 | −8.5 | 35.9 | 13.8 |
| 2015 | US15 | −2.8 | −1.1 | −5.9 | −2.2 | 8.7 | 3.7 |
| 50 year | US50-RCP4.5 | −13.8 | −3.7 | −34.8 | −8 | 36.3 | 13.0 |
| 100 year | US100-RCP4.5 | −6.4 | −2.1 | −15.2 | −4.3 | 17.3 | 6.5 |
| 50 year | US50-RCP8.5 | −5.8 | −2.1 | −13.1 | −4.4 | 17.3 | 6.8 |
| 100 year | US100-RCP8.5 | −2.7 | −1.0 | −5.9 | −2.1 | 7.6 | 3.0 |

### 3.1.2. Impact of Dredging the Tanks

Figures 8–14 show that decrease in flood peaks is higher in the 50-year return period, i.e., 2005 flood events when compared to the 100-year return period or 2015-like flood events. Additionally, Table 4 shows that the 2-m dredging scenarios exhibit higher peak reduction in the case of moderate flood events like the 50-year return period or 2005-flood-like rainfall scenarios. From Figure 8a, it can be noticed that the peak discharge of a 2005-like flood for the US05-TP2m-F50 scenario is closer to that for H05-TP-F50 scenario. In other words, for moderate 2005-flood-like scenarios with lesser initial storage in the tanks, dredging them by 2 m may neutralize the impact of urbanization on flooding. However, such moderation is not observed for F75 and 2015-like flood scenarios. Thus, as mentioned earlier, dredging of the tanks will be more effective when initial storage in the tanks is moderated to less than 50% prior to a flooding event.

### 3.1.3. Impact of the Disappearance of Tanks

The peak flows upstream of CMA for the H05-TP-F50 and H15-TP-F50 scenarios (Figures 8a and 9a) are found to be 1786 $m^3/s$ and 3267 $m^3/s$, respectively. However, if the 2005 and 2015 floods were to occur in 2050 and the tanks are to disappear, i.e., for the US-TD scenario (Figure 14a,b), the peak flows will increase to 3340 $m^3/s$ (an increase by 1554 $m^3/s$) and 4029 $m^3/s$ (an increase by 762 $m^3/s$), respectively. It can also be noticed from Figures 9a and 14a that the peak discharge for 2005 flood in the US05-TD scenario is closer to that of the 2015 flood in H15-TP scenarios. In other words, if the tanks in the upstream portion of the basin lose their storage completely in the future, coupled with urban sprawl, then even a moderate flood like the one that occurred in 2005 will behave like the extreme flood that occurred in 2015. Importantly, Figure 14 and Table 4 show that the effect of dredging and disappearance of tanks on peak flows is more significant for moderate floods such as that occurred in 2005 as compared to extreme floods that occurred in 2015. Peak flows for the 24-h duration 100-year return period rainfall BL100-TP-F50 and BL100-TP-F50 scenarios are found to be 1705 $m^3/s$ and 2954 $m^3/s$, respectively. There is an increase in the peak flow by 295 $m^3/s$ and 226 $m^3/s$ for the US100-RCP4.5-TD and US100-RCP8.5-TD, respectively, from the corresponding US100-RCP4.5-TP-F50/US100-RCP8.5-TP-F50 scenarios. Similarly, for the 24-h duration 50-year return period flood, the peak flows are increased by 528 $m^3/s$ and 463 $m^3/s$ for the US50-RCP4.5-TD and US50-RCP8.5-TD scenarios, respectively, from the corresponding TP-F50 scenarios. As one would expect, the peak flows and flood extents of 24-h duration RCP scenarios are less than that of actual 2005 or 2015 flood events. This can be attributed to the fact that, in the hydrologic simulations with the NEX-GDDP projected rainfall (design storm rainfall) from RCP, we did not account for pre-event rainfall (antecedent conditions), which plays an important role in changing the initial storage in a basin. Also, it has to be noted that only 24-h duration rainfall events are considered herein for future design floods.

### 3.2. Hydraulic Results

The effect of tanks in flood moderation in CMA needs to be looked at in terms of inundation depths and extents for a holistic understanding of flood hazards.

### 3.2.1. Analysis of Inundation Depth

Important places (Figure 15) that are located on the banks and floodplains of Adyar River are chosen for comparison of the inundation depths of various scenarios (Figure 15). It has to be mentioned that only US50-RCP8.5 scenarios are presented in the paper as they represent the worst-case possibilities. Irrespective of the initial storage levels in the tanks, dredging of tanks by 2 m seems to reduce the inundation depths considerably for moderate floods such as 2005 flood and 50-year return period floods, when compared to those for extreme events such as 2015 flood and 100-year return period floods (Figures 16–19). For the US05-TP2m-F75 scenario, the inundation depths at the chosen locations are lesser when compared to that of the H05-TP-F75 scenario. In other words, it can be said, that if the 2005 and similar flood events occur in 2050 on an urbanized domain that has its tanks preserved and dredged by

2 m, then irrespective of the initial storage in the tanks, the effect of urbanization on flooding will be almost neutralized. However, as anticipated, the reduction in inundation depth is greater for 50% initial storage with 2-m dredging scenarios. In addition, the impact of dredging of tanks on flood moderation is reflected more in the case of moderate 2005-flood-like scenarios. On the other hand, for the extreme 2015-flood-like scenarios, dredging will not result in appreciable reduction of inundation depths. In all scenarios, the reduction in inundation depths is seen to exhibit a decreasing trend from upstream to downstream of the domain. This can be attributed to the fact that the areas in the downstream portion are closer to sea level and are relatively flat.

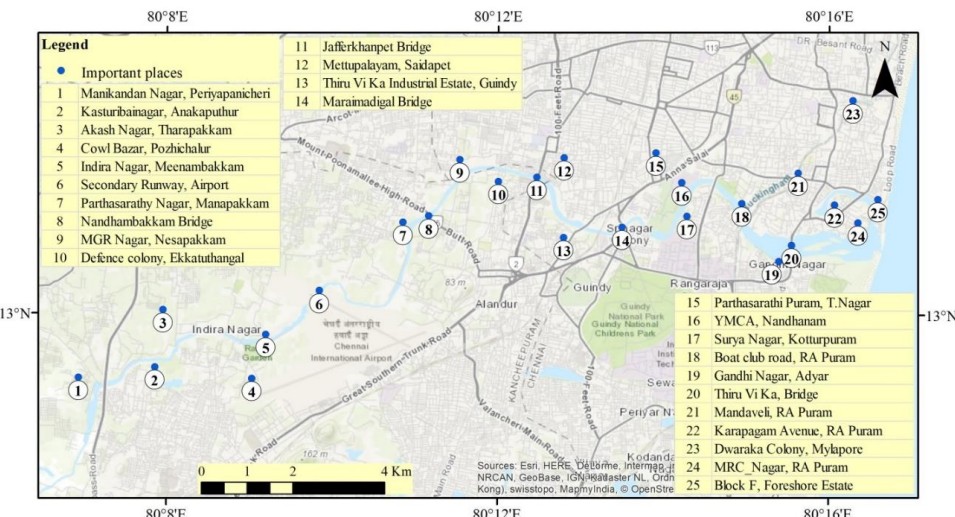

**Figure 15.** Location of important places in the study area.

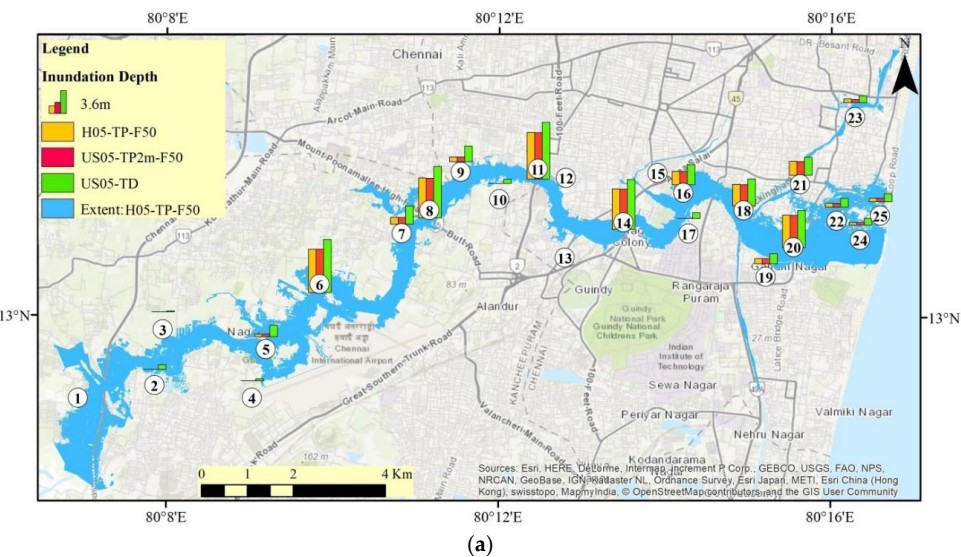

**(a)**

**Figure 16.** *Cont.*

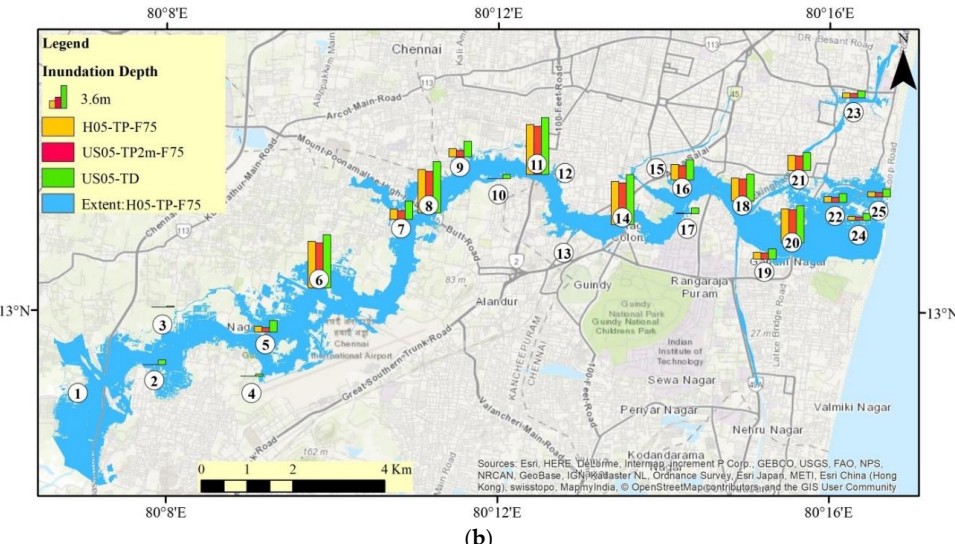

**Figure 16.** Depth comparisons for 2005 flood scenarios: (**a**) 50% and (**b**) 75% initial storage in the tanks.

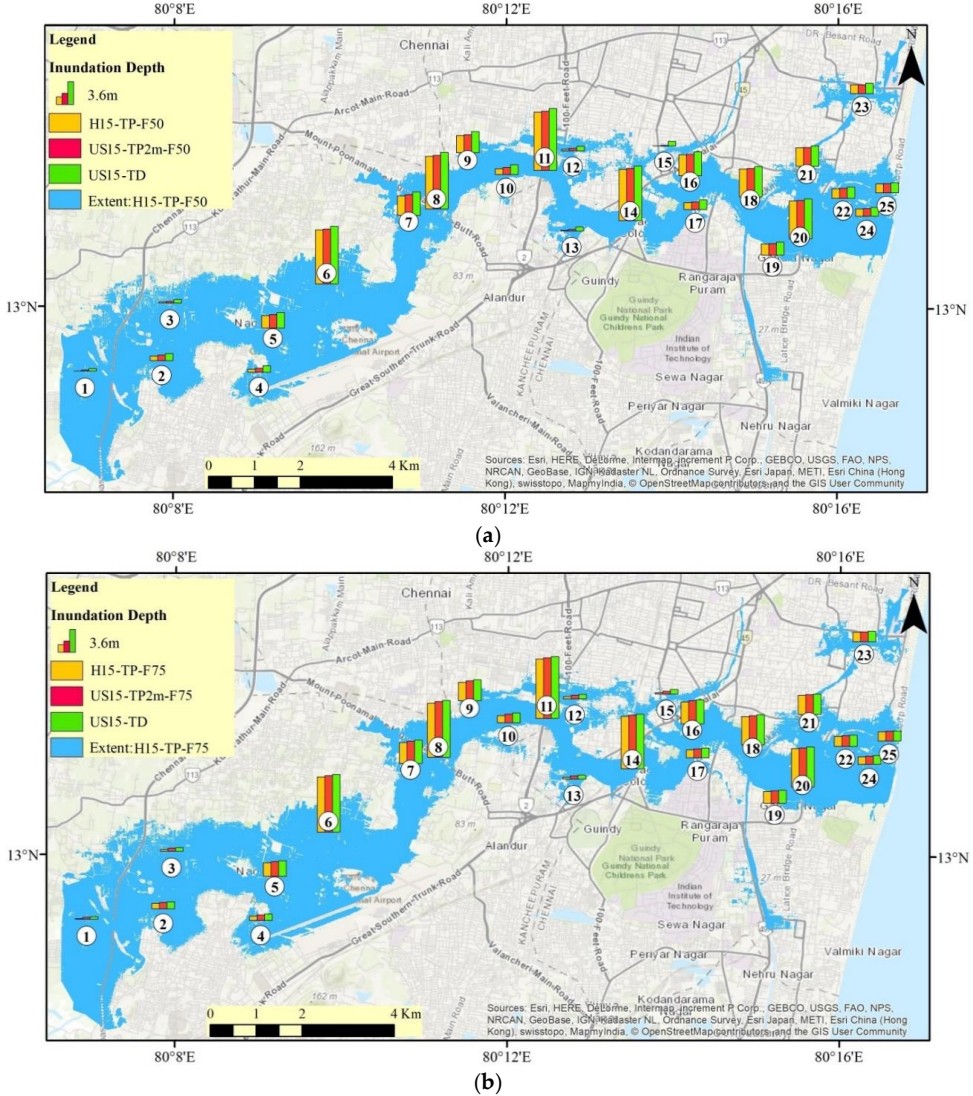

**Figure 17.** Depth comparisons for 2015 flood scenarios: (**a**) 50% and (**b**) 75% initial storage in the tanks.

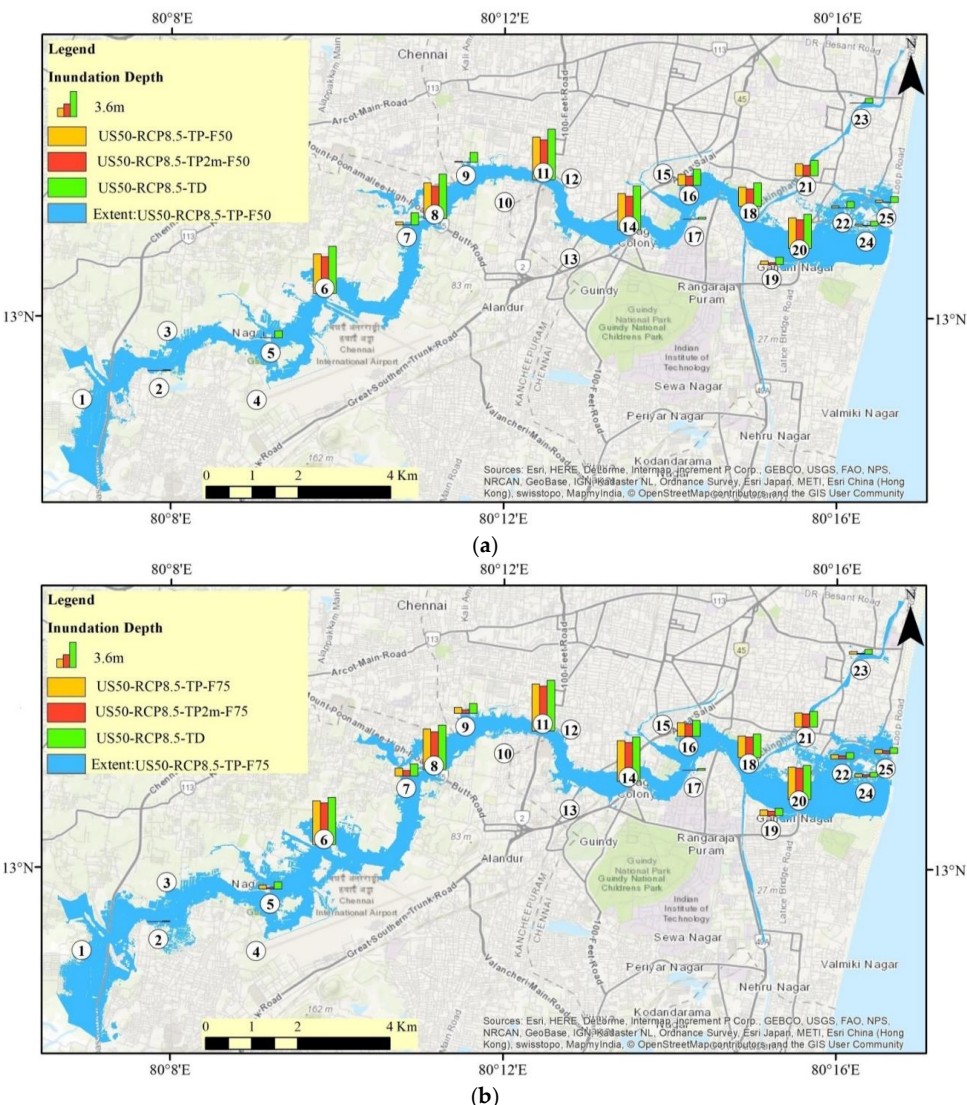

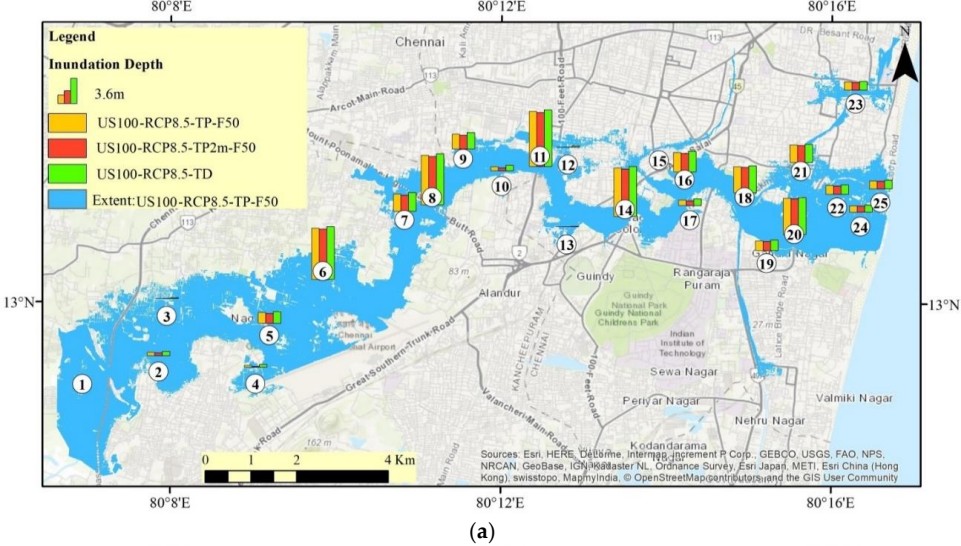

**Figure 18.** Depth comparisons for the 50-year return period and CNRM 8.5 flood scenarios for (**a**) 50% and (**b**) 75% initial storage in the tanks.

**Figure 19.** *Cont.*

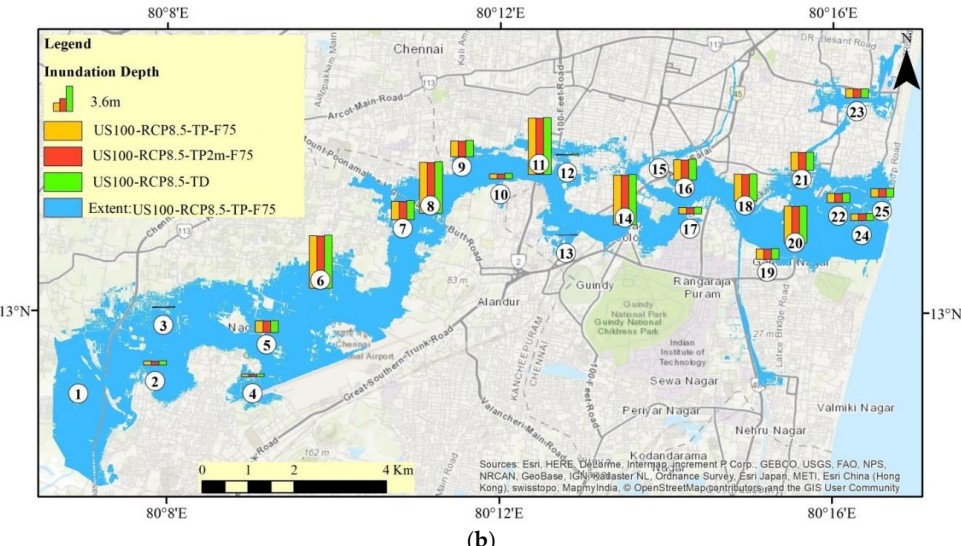

(**b**)

**Figure 19.** Depth comparisons for the 100-year return period and CNRM 8.5 flood scenarios for (**a**) 50% and (**b**) 75% initial storage in the tanks.

### 3.2.2. Analysis of Inundation Extent

Figure 20 shows inundation extents for various 2005- and 2015-flood-like scenarios and US50/100-RCP8.5 scenarios. From Figure 20c,f, it can be seen that the US15-TP2m-F50 and US15-TD scenarios do not show noticeable differences in flooding extents for extreme flooding scenarios. On the other hand, in the case of moderate floods, US05-TP2m-F50 exhibits a flooding pattern similar to that of the H05-TP-F50 scenarios (Figure 16, Figure 18, and Figure 20a,b). Thus, in terms of inundation extents, dredging of the tanks by 2 m can neutralize the effect of urbanization for a moderate 2005-flood-like scenario. In the case of disappearance of tanks and urban expansion, a greater increase in inundation areas with respect to the H05-TP-F50 scenario is observed for the 2005-flood-like scenario than the H15-TP-F50 2015-flood-like scenario. Nevertheless, for moderate 50-year return period floods or 2005-flood-like scenarios, conservation and dredging of tanks results in reduction in inundation extents irrespective of the initial storage in the tanks (Figure 20a,b,d,e). Inundation extents for TB2m-F75 and tank disappearance scenarios for extreme floods did not show noticeable differences, and hence, the figures for the same are not provided in the paper. The inundation pattern of the H05-TD scenario is similar to that of the H15-TP-F50 scenario (Figures 17a and 20a). In other words, in the future, a complete loss of storage of tanks in the upstream may cause a moderate 2005-flood-like event (50-year return period rainfall) to inundate to the scale of the extreme 2015-flood-like event (100-year return period rainfall). As a consequence, this will result in an excessively high inundation, especially in the areas (Figure 20a) closer and upstream of Chennai International Airport and those that are closer to the Adyar creek and Mambalam drain. On the other hand, different 2015-flood-like scenarios exhibit only marginal differences in the inundation extents (Figure 20c,e). It is noteworthy to mention that similar inferences are drawn in the previous section.

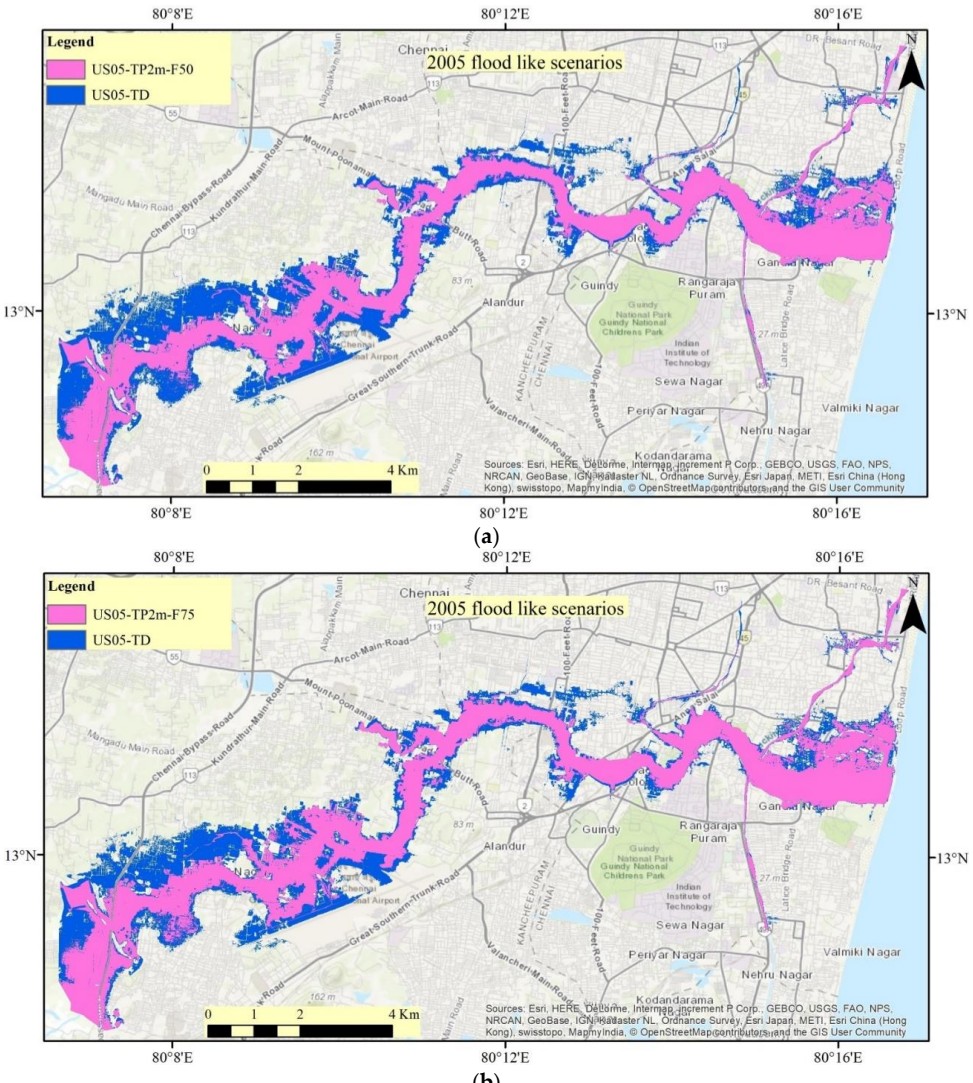

(**a**)

(**b**)

**Figure 20.** *Cont.*

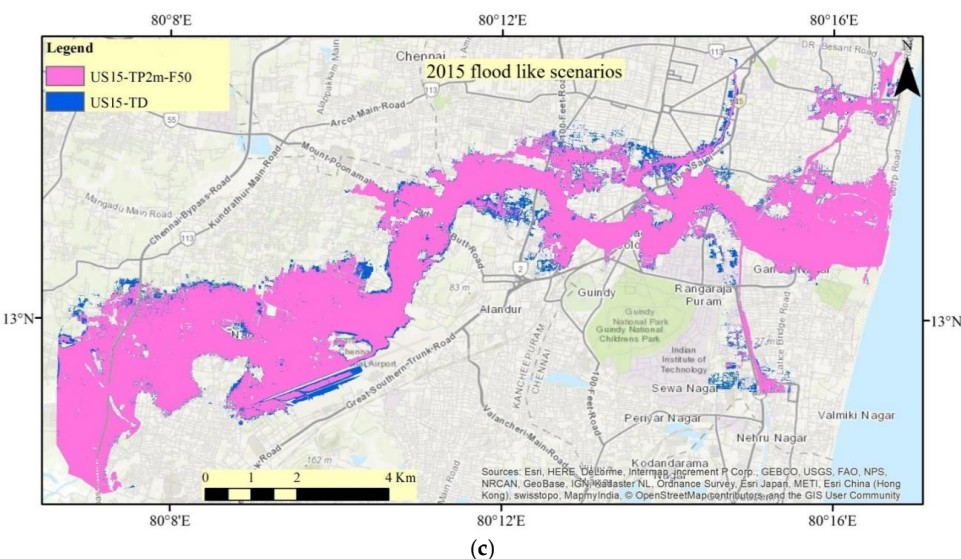

(**c**)

**Figure 20.** *Cont.*

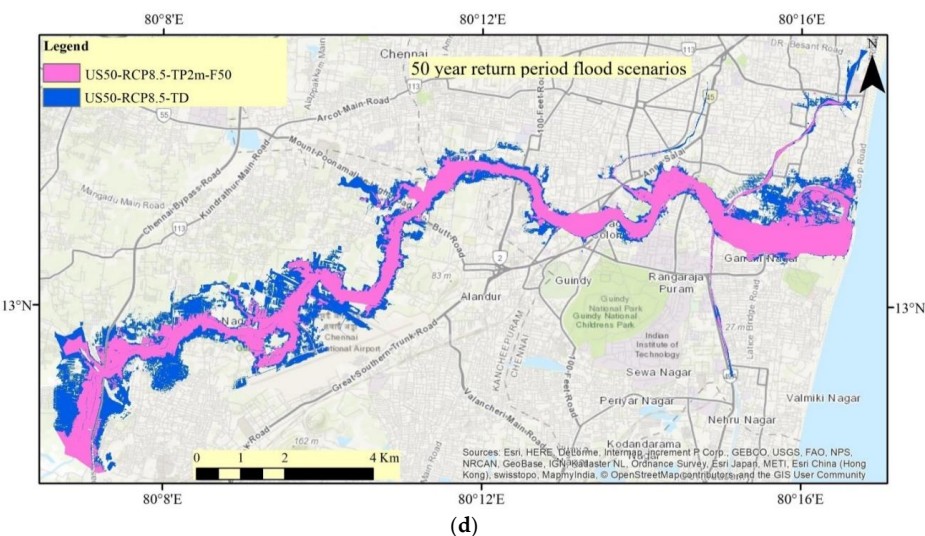

(**d**)

**Figure 20.** *Cont.*

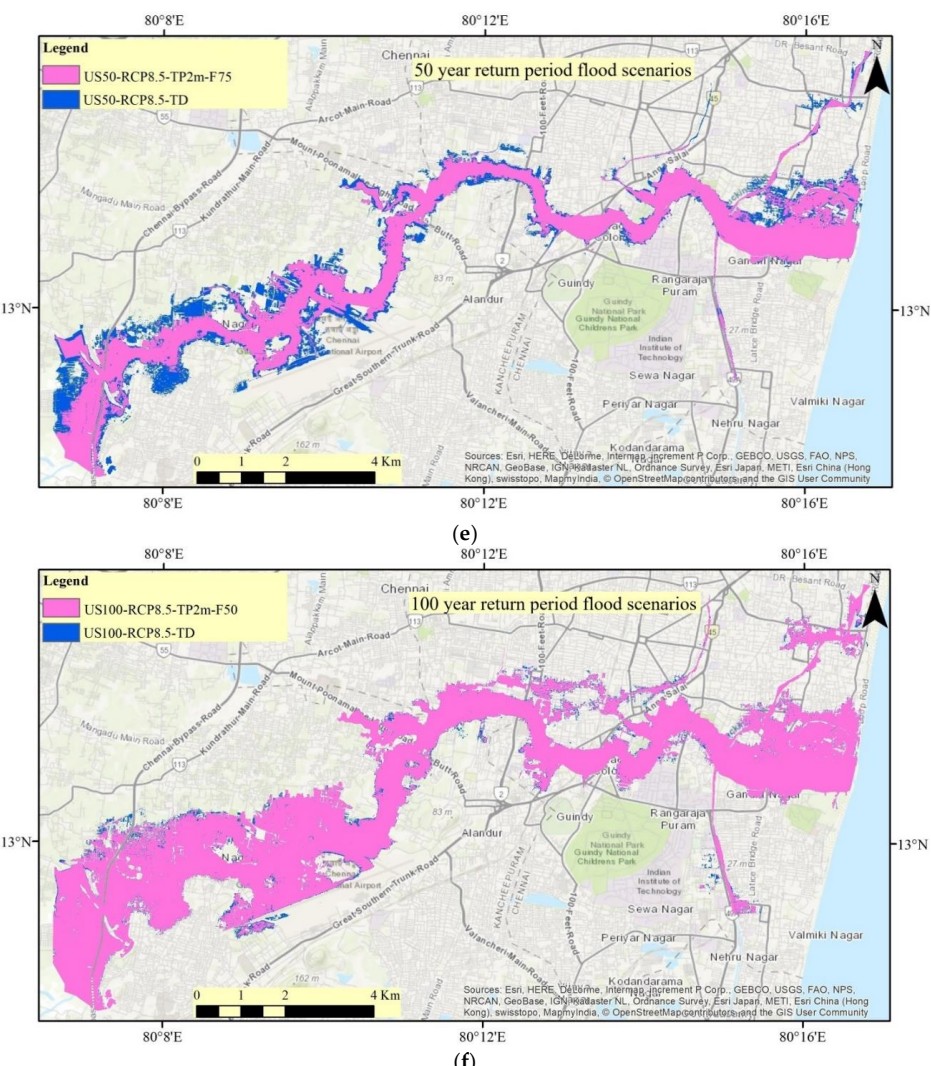

(**e**)

(**f**)

**Figure 20.** Simulated inundation extents for the scenarios: 2005 flood event: (**a**) fill50 and (**b**) fill75; 2015 flood event: (**c**) fill50; climate change scenarios (CNRM 8.5): (**d**) fill50 (50-year return period), (**e**) fill75 (50-year return period), and (**f**) fill50 (100-year return period).

Findings on relative percentage changes in the total inundation area for various scenarios are reported in Table 5. For the 2005 flood, the total area under inundation increases from 18.6 km$^2$ to 27.8 km$^2$ as one moves from the historical existing scenario to the future (2050) urban sprawl and disappearance of tanks scenarios. Similarly, in the case of the 2015 flood, the area of inundation increases from 33 km$^2$ to 40.3 km$^2$. From Table 5 and Figure 20, it can be seen that, in the combined urbanization and future climate F50 scenarios, the dredging and disappearance of tanks have a greater impact on the inundation areas for 50-year return period floods in comparison to 100-year return period floods. However, dredging–US05-TP2m-F75 scenarios show greater reduction in inundation areas (27.4%) with respect to US05-TP-F75 in comparison to the F50 scenarios, unlike the trend in peak flows (−8.5%). The disappearance of tanks has caused an increase in inundation areas by 4 km$^2$ and 4.8 km$^2$ relative to the US50-RCP4.5-F50 and US50-RCP8.5-F50 scenarios, respectively, for the 50-year return period rainfall. In the case of 100-year rainfall, the disappearance of tanks resulted in an increase in inundation areas by 2.5 km$^2$ and 2.4 km$^2$ relative to the US100-RCP4.5-F50 and US100-RCP8.5-F50 scenarios, respectively. As seen earlier, a similar trend is reported in the peak flows (Tables 4 and 5). This reiterates the fact that tanks play a crucial role in improving the damages caused by floods, especially in the case of a moderate, one in 50-year flood events.

**Table 5.** Relative percentage change in inundation area for various scenarios in comparison to the corresponding existing scenarios of urban sprawl and climate change.

| Rainfall | Scenarios | TP1m | | TP2m | | TD | |
|---|---|---|---|---|---|---|---|
| | | F50 | F75 | F50 | F75 | F50 | F75 |
| 2005 | US05 | −11.2 | −3.6 | −20.2 | −27.4 | 19.3 | 1.5 |
| 2015 | US15 | −3.0 | −0.8 | −6.0 | −2.1 | 9.5 | 4.1 |
| 50 year | US50-RCP4.5 | −7.9 | −3.2 | −15.2 | −5.9 | 24.4 | 10.3 |
| 100 year | US100-RCP4.5 | −4.4 | −1.5 | −9.2 | −3.5 | 13.6 | 5.6 |
| 50 year | US50-RCP8.5 | −6.4 | −2.1 | −12.8 | −4.6 | 18.0 | 7.5 |
| 100 year | US100-RCP8.5 | −3.0 | −0.9 | −5.9 | −1.9 | 7.9 | 3.1 |

### 3.2.3. Flood Hazard Zonation: Impact Analysis

The inundation depths 0.6 m, 1.4 m, and 3.5 m are chosen as threshold depths for defining flood hazard zones (Table 6) based on the satellite- and field-based damage for 2015 [37,64]. Higher differences in the inundation area are observed under moderate hazard zones across various scenarios for the 2015 flood. However, for the 2005 flood, higher differences are observed under low hazard zones across all scenarios. In the case of moderate rainfall events like 2005/RCP 4.5-flood-like events, the inundated area under "very high" hazard category is more when compared to the "low" category. It is the reverse for extreme 2015/RCP 8.5-flood-like events. This is due to the fact that, in the 2005 flood, the floodplains were inundated minimally, unlike the one in hundred-year 2015 flood. However, Figure 21 shows that, in all scenarios, the area under the high hazard zone category is the highest. It can also be seen from Figure 21 that the reduction in total and hazard zone-wise inundation areas is greater for lesser initial storage in the tanks across all the flood scenarios. This inference is consistent with the results reported in the previous sections.

**Table 6.** Details of the flood hazard zones used in the study.

| Hazard Zone | Depth of Inundation (m) | Description |
|---|---|---|
| Low | 0.1 to 0.6 | Flood waters enter into the building. |
| Moderate | 0.6 to 1.4 | Half of the ground floor gets inundated up to the height of furniture; the cars get nearly submerged. |
| High | 1.4 to 3.5 | Ground floor of the building gets submerged, and flood waters may enter in to the first floor. |
| Very High | >3.5 | First floor of the building also gets affected. |

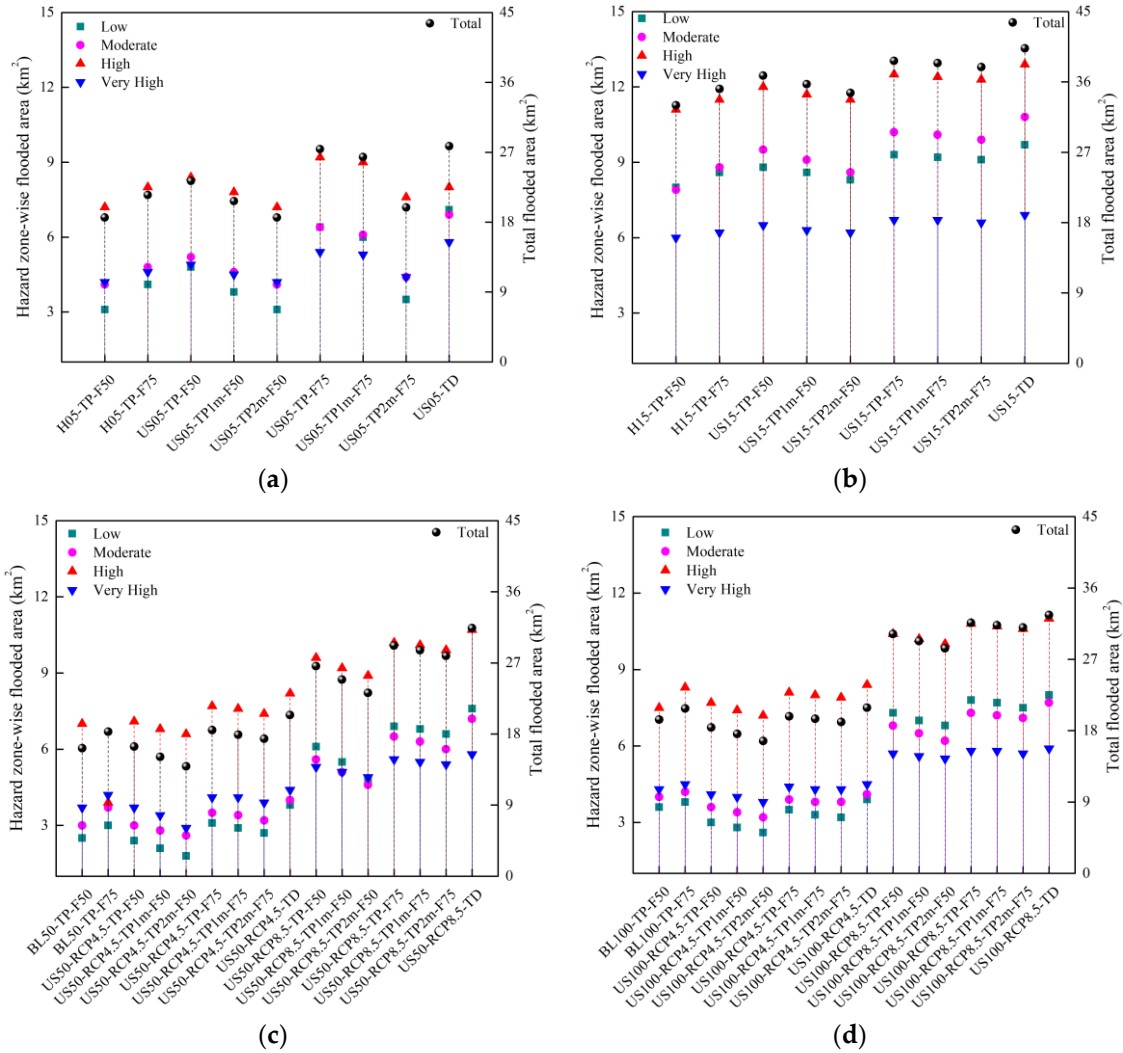

**Figure 21.** Area under various hazard zones for the flood events: (**a**) 2005, (**b**) 2015, and climate change scenarios (CNRM 4.5 and 8.5) for the (**c**) 50-year and (**d**) 100-year return period flood scenarios.

## 4. Limitations of the Present Study

It has to be borne in mind that the future climate scenarios used in this study do not account for impact of urbanization on the urban microclimate. The tide gauge station data analysis shows a sea level rise at a mere rate of 0.7 mm/year (data available at http://www.psmsl.org/data/obtaining/stations/205.php). Therefore, sea level rise by 2050 is not accounted for in this study. Due to lack of data on storm drains and their status of functioning, the study considers only flooding in the vicinity of Adyar River and not the widespread pluvial flooding. To get a holistic view of the role of tanks in flood moderation, different return period floods of different duration and climate models are to be considered. Nevertheless, in the study area, the extreme NEMR usually lasts for 2–3 days and these events may occur in combination with a lesser or heavier pre-event spell. Therefore, the cases "what if a historical extreme event happens in the future that entails urban expansion" and "what if the climate projected 24-h duration extreme event occurs in the future that entails urban expansion" have been considered in this study. Moreover, from Figure 14, it can be seen that the negative impact of disappearance of tanks is reflected more in event-based simulations in comparison to that of the climate projected 24-h duration rainfall. Nevertheless, this study is highly valuable as it provides the necessary scientific reasons for preservation and maintenance of the existing tanks at the very

least. Further studies need to be carried out to evaluate other associated benefits of the same, such as increased ground water recharge, water availability for domestic supply, and irrigation.

## 5. Summary and Conclusions

Chennai city is prone to frequent flooding due to deep depression and cyclonic activities during the NE monsoon. In order to alleviate fluvial flooding in the Adyar basin that comprises the southern part of the CMA, the effect of dredging of 163 existing tanks under urbanization and various rainfall scenarios are analyzed in this study. The Adyar River flowing through the CMA functions as a surplus course for the tanks. The impact of dredging of tanks by 1 m and 2 m on the flooding in CMA is investigated by simulating a moderate 1 in 50-year flood, i.e., a 2005-flood-like event, and an extreme 1 in 100-year flood, i.e., a 2015-flood-like event. The extreme rainfall events for the mid-century period obtained from the CNRM models are also simulated. The study is based on the urbanization-hydrologic-hydraulic modelling framework under various rainfall scenarios and brings out the importance of maintaining and dredging of the tanks in flood moderation of CMA. The major conclusions from this study that are specific to the Adyar basin are as follows:

(i)  It can be concluded that the tanks in the upstream catchments provide effective flood mitigation for a downstream city, when the initial storage in the tanks is minimum or when there is an insignificant wet spell prior to occurrence of the extreme rainfall. Hence, an accurate short-range rainfall forecast system for the Adyar basin can help the authorities in charge of the tanks release excess water in the tanks to downstream areas or store it in secondary reservoirs on time.

(ii)  The dredging of tanks results in considerable differences in inundation depths and extents, irrespective of the 50% and 75% initial storage for moderate 2005-flood-like events, i.e., one in 50-year return period floods.

(iii)  Dredging of tanks uniformly by 2 m within the tanks' water spread area does not reduce inundation significantly for 1 in 100-year flood-like scenarios. However, a similar dredging of the tanks can neutralize the adverse effect of urbanization in 2050 in the case of 1 in 50-year flood-like scenarios.

(iv)  In the future, if the tanks disappear due to urbanization, then even for a moderate 1 in 50-year rainfall, there will be excessive inundation due to fluvial flooding closer to that caused by a 1 in 100-year rainfall, i.e., similar to the 2015 flood event.

(v)  As the Adyar basin is prone to occurrence of frequent extreme events, it is important that the uncontrolled tanks are equipped with flood gates to regulate storage prior to a forecasted flooding event to minimize flood peaks.

(vi)  The flood hazard analysis performed in this study indicates that higher differences in the inundation area due to dredging and disappearance of tanks get reflected under the low hazard zone category for 1 in 50-year flood-like events. On the other hand, the differences are greater in moderate zones for 1 in 100-year flood-like events. Such a trend is due to the fact that the 1 in 100-year floods are greater in magnitude than the 1 in 50-year floods. Hence, even though the reduction/increase in inundation area due to dredging/disappearance of tanks is less for 1 in 100-year floods, the implications of dredging of tanks in flood management cannot be discounted.

(vii)  This study provides insights into dredging of tanks by 1 m and 2 m as possible flood mitigation scenarios. The observations of this study can be useful to the government for the proposed plan of dredging of tanks in the Adyar basin.

It is obvious that dredging is a very expensive process. However, only a very few options are available for flood mitigation in the CMA owing to the dense population, space constraint for constructing hydraulic structures, and the almost flat terrain of CMA. Therefore, the key takeaway from this study is that, at the very least, urban planning should consider preserving the storage and functionality (maintenance of tank bund and feeder channels) of the existing tanks so as to minimize damage from moderate fluvial floods (1 in 50-year rainfall-like events) in the Adyar basin in the future,

given the reality that urban development cannot be efficiently controlled in a developing economy. The aforementioned inferences from the study can be used as guidelines for the government for better preparedness and management of floods in CMA.

**Author Contributions:** Conceptualization, N.N.D. and S.N.K.; methodology, N.N.D., B.S., V.M.B., B.N., S.M.B., and S.N.K.; software, N.N.D. and B.S.; validation, N.N.D., B.S., V.M.B., B.N., S.M.B., and S.N.K.; formal analysis, N.N.D., B.N., S.M.B., and S.N.K.; investigation, N.N.D., B.S., and S.N.K.; resources, B.N., S.M.B., and S.N.K.; data curation, B.N., B.S., C.M.B., T.U., and D.T.V.; writing—original draft preparation, N.N.D. and S.N.K.; writing—review and editing, S.N.K., B.N., and B.S.; supervision, S.N.K.; funding acquisition, S.N.K. All authors have read and agreed to the published version of the manuscript.

**Funding:** This research was funded by the Department of Science and Technology, India under SPLICE–Climate Change Programme (grant number: DST/CCP/CoE/141/2018C).

**Acknowledgments:** The authors are thankful for the support provided by the Department of Science and Technology, India under SPLICE–Climate Change Programme. The authors are also thankful to a number of institutions for collecting and sharing relevant data to carry out this study. This data collection was supported by the Office of the Principle Scientific Advisor, Government of India (sanction order: Prn.SA/AB/Urban Floods/2016(G)).

**Conflicts of Interest:** The authors declare no conflict of interest.

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
