# Peer review of "Investigation of Role of Retention Storage in Tanks (Small Water Bodies) on Future Urban Flooding: A Case Study of Chennai City, India"

_water, doi:10.3390/w12102875_

Round 1

Reviewer 1 Report

This paper investigates the effects of retention or detention tanks on the flooding control and management. An example of Chennai City has been taken for illustration of the developed application procedure. 

Overall, this paper falls within the scope of this journal, and the topic may be of great interests to readers and significance to the community (practice and research). But a revision is required for this paper publication on the Water Journal. Key issues are as follows:

(1) I would suggest the title change to "Investigation of the role of retention tanks on urban flooding management: A case study of Chennai City, India", which can precisely reflect the objectives and contents of current paper work;

(2) The literature review part should be revised, to sharply focus on the field of research topic. Specifically, two key words: "Retention or Detention Tanks" & "Urban Flooding"; Especially, the authors are recommend to review and cite more recent papers, such as those listed in the end of this review (including those published on this Water journal, see comment (7) below).

(3) In the introduction part, a brief discussion on the previous studies, in particular for those very related to current paper, so as to make clear on the different of current paper from those, with highlighting the contributions of current work. This will be very important to be a technical paper;

(4) Figure 1 is important to describe the methodology of this study, but it in current form is too complicated to read. It can be divided into several figures with different scales/frameworks, or improve it in other ways;

(5) The results of this paper look very good and the data evidence is strong to draw those conclusions in the paper. Also the authors are appreciated to present the limitations of this study, which will be useful to the future study in this field; This is a comment (rather than an issue);

(6) The conclusion part can be shortened to list the key findings only.

(7) the reference style is not consistent with the journal requirement, and the authors are suggested to adjust them accordingly. More relevant references can be found with following paper DOIs: 10.2166/ws.2008.029; 10.3390/w11071389; 10.1007/s11269-015-0931-0; 10.1007/s11269-016-1282-1; 10.1007/s11269-016-1444-1; 10.1007/s11269-019-02300-0; 10.1016/j.jenvman.2012.02.003; 10.1061/(ASCE)IR.1943-4774.0000927; etc.

In summary, this paper is in good quality for its research objective, method and results. A revision on addressing above issues should be made prior to its publication. Look forward to reviewing the revised version.

Author Response

Dear Reviewer,

We would like to thank you for your valuable comments and observations. We have addressed all the comments and they are attached in the file. The minor corrections are not shown in the track change version.

with regards,

Soumendra

Reviewer 2 Report

Tthe manuscript is very interesting and deals with a current and very sensitive issue at this time of climate change. It could be an excellent planning tool, and the methodology used could also be exported to other areas of the world to plan a plan.

You need to correct some minor errors:

Line 35-36-37: What is the reference (indicate it).

Line 118-123: remove all, they are advices.

Line 124-136: not appropriate under the paragraph Methodology, create a sub-paragraph to enter these informations.

Line 183: remove the full stop.

Author Response

Dear Reviewer,

We are extremely thankful to your constructive comments. We have addressed all the comments. The minor changes are not shown in the track change version. The replies to your comments are attached.

with regards,

Soumendra

Round 2

Reviewer 1 Report

The authors have addressed most of my comments. For some other minor issues such as typos and writing as well as format, I believe these could be resolved by the proofreading process. Now I would like to suggest to accept this paper.